# Effects of Nicotinamide Mononucleotide Supplementation and Aerobic Exercise on Metabolic Health and Physical Performance in Aged Mice

**DOI:** 10.3390/nu17193148

**Published:** 2025-10-02

**Authors:** Yi-Ju Hsu, Mon-Chien Lee, Huai-Yu Fan, Yu-Ching Lo

**Affiliations:** Graduate Institute of Sports Science, National Taiwan Sport University, Taoyuan 33301, Taiwan; kurt0710@ntsu.edu.tw (M.-C.L.); 1130211@ntsu.edu.tw (H.-Y.F.); 1110206@ntsu.edu.tw (Y.-C.L.)

**Keywords:** aging, aerobic exercise, nicotinamide mononucleotide (NMN), NAMPT, energy metabolism, exercise performance

## Abstract

**Background/Objectives**: Aging is characterized by progressive physiological and metabolic decline. Aerobic exercise mitigates age-related impairments, and nicotinamide mononucleotide (NMN), a precursor in the NAD^+^ salvage pathway, has emerged as a nutritional intervention to promote healthy aging. This study investigated whether NMN supplementation combined with aerobic exercise provides synergistic benefits on physical performance and metabolic regulation in aged mice. **Methods**: Forty male C57BL/6J mice, including eight young (8 weeks) and thirty-two aged (85 weeks) mice, were randomly assigned to five groups: young sedentary (YS), aged sedentary (AS), aged with exercise (AE), aged with NMN (ASNMN; 300 mg/kg/day), and aged with combined NMN and exercise (AENMN). Interventions lasted six weeks. Assessments included grip strength, muscle endurance, aerobic capacity, oral glucose tolerance test (OGTT), and indirect calorimetry, followed by biochemical and molecular analyses of NAMPT and SirT1 expression. **Results**: The AENMN group demonstrated significant improvements in maximal strength and aerobic endurance compared with the AS group (*p* < 0.05). Both NMN and exercise interventions increased blood NAMPT concentrations, with the highest levels observed in the AENMN group (*p* < 0.05). SirT1 expression was elevated in the ASNMN and AENMN groups relative to YS (*p* < 0.05). Glucose tolerance improved in the ASNMN and AENMN groups (*p* < 0.05). Enhanced energy metabolism in the AENMN group was indicated by increased oxygen consumption, elevated energy expenditure, and reduced respiratory quotient. **Conclusions**: NMN supplementation, particularly when combined with aerobic exercise, effectively improved aerobic performance, glucose regulation, and systemic energy metabolism in aged mice. These findings suggest that NMN, in synergy with exercise, may serve as a promising nutritional strategy to counteract age-associated metabolic and functional decline.

## 1. Introduction

Aging is an inevitable and irreversible biological process characterized by the progressive accumulation of molecular and cellular damage from intrinsic and extrinsic stressors over time [1]. This damage leads to a gradual decline in the function of organ systems and increases vulnerability to disease. The aging process involves complex, interrelated mechanisms, including genomic instability, mitochondrial dysfunction, telomere attrition, epigenetic alterations, dysregulated nutrient sensing, loss of proteostasis, cellular senescence, stem cell exhaustion, altered intercellular communication, and chronic oxidative stress [2]. In skeletal muscle, aging is associated with mitochondrial dysfunction and impaired substrate utilization, which contributes to lipid accumulation, insulin resistance, and metabolic disorders [3]. These pathophysiological changes increase the risk of multiple chronic conditions such as metabolic syndrome, cardiovascular disease, neurodegenerative disorders, osteoporosis, sarcopenia, and frailty [4,5]. Given the increasing burden of multimorbidity in aging populations, effective strategies for delaying or mitigating age-associated functional decline are of urgent clinical relevance. Among the proposed interventions, appropriate exercise, dietary modification, targeted supplementation, and pharmacological agents have shown potential for slowing aging processes [6]. Aging phenotypes involve complex and multifactorial interactions, and further research is needed to identify integrative and effective anti-aging strategies [2,7].

Regular physical activity is widely recognized as one of the most effective non-pharmacological interventions to counteract aging and extend health span [8]. Exercise promotes neurogenesis and protects against neurodegeneration, improves cardiovascular function, enhances metabolic regulation, and helps to maintain muscle mass, strength, and functional capacity in older adults [9]. In addition to its systemic benefits, exercise training upregulates nicotinamide adenine dinucleotide (NAD^+^) levels and related metabolic enzymes in multiple tissues, thereby counteracting the age-associated decline in NAD^+^ availability [10]. NAD^+^ is an essential coenzyme that supports cellular redox reactions and energy metabolism through glycolysis, oxidative phosphorylation, and ATP production [11]. Structurally, NAD^+^ comprises adenosine 5′-phosphate and nicotinamide ribose 5′-phosphate linked by a pyrophosphate bridge [12]. Beyond its role in redox balance, NAD^+^ functions as a critical substrate for over 500 enzymatic reactions, supporting diverse biological processes such as mitochondrial biogenesis, DNA repair, chromatin remodeling, inflammation control, and stem cell maintenance [13,14]. These functions make NAD^+^ a central regulator of cellular health and a key molecular target in aging research [12,15]. Lifestyle interventions such as caloric restriction, intermittent fasting, thermal stress, and carbohydrate limitation have also been shown to boost NAD^+^ levels [16]. Moreover, NAD^+^ precursors like nicotinamide mononucleotide (NMN) and nicotinamide riboside (NR) have attracted attention due to their ability to restore NAD^+^ biosynthesis and combat metabolic decline during aging [17].

NMN is a critical intermediate in the NAD^+^ salvage pathway. It is synthesized from nicotinamide (NAM) and 5-phosphoribosyl-1-pyrophosphate (PRPP) through the catalytic action of nicotinamide phosphoribosyltransferase (NAMPT) [18]. NMN exists in two isomeric forms (alpha and beta isomers) that are biologically active and naturally found in vegetables, fruits, and meats [19]. Following oral intake, NMN is absorbed primarily in the intestine where it undergoes dephosphorylation to form nicotinamide riboside (NR). NR then re-enters the NAD^+^ biosynthetic pathway by phosphorylation through nicotinamide riboside kinase (NRK) and is subsequently converted into NAD^+^ by NMN adenylyltransferase (NMNAT) in an ATP-dependent reaction [20]. This metabolic conversion enables NMN to rapidly elevate NAD^+^ levels in vivo and support a variety of biological processes. These include the regulation of energy metabolism, chromatin remodeling, circadian rhythm, DNA repair, and immune responses [21]. The ability of NMN to restore cellular NAD^+^ levels has attracted increasing interest in the field of aging research. Studies have shown that NMN supplementation attenuates functional decline and protects against metabolic dysfunctions during aging [22]. Dietary intake of NMN or related NAD^+^ precursors also increase NAD^+^ concentrations in the blood of middle-aged and elderly individuals [23,24]. A recent randomized controlled trial revealed that daily supplementation with 250 mg NMN for 12 weeks significantly improved gait speed, handgrip strength, and lower-limb functional performance in older adults [25]. In addition, long-term NMN administration in mice restores hepatic NAD^+^ levels and reverses age-related transcriptomic changes in liver tissue [26]. These findings provide compelling evidence that NMN is not only a metabolic precursor, but also a promising nutritional agent with physiological relevance to aging.

Endurance exercise has long been recognized as an effective strategy to promote mitochondrial biogenesis, improve vascular endothelial function, and enhance aerobic capacity, particularly among older populations [27]. Interestingly, NMN supplementation induces comparable physiological adaptations, including improvements in mitochondrial efficiency, oxygen utilization, and skeletal muscle capillary density [28]. These overlapping benefits suggest that NMN and aerobic exercise may act through shared or complementary mechanisms that support systemic health during aging.

Therefore, this study aimed to evaluate whether the combination of NMN supplementation and endurance exercise could provide additive or synergistic benefits in mitigating age-related dysfunction. This study focused on evaluating aging-related changes in energy metabolism, body composition, and physical performance in naturally aged mice, with particular attention given to adaptations in mitochondrial function and glucose regulation that are commonly impaired during aging. We further examined whether this combined intervention could enhance the NAD^+^ salvage pathway and contribute to the maintenance of metabolic homeostasis, with the goal of mitigating age-related functional deterioration more effectively than NMN or exercise alone.

## 2. Materials and Methods

### 2.1. Experimental Design

All animal experiments were reviewed and approved by the Institutional Animal Care and Use Committee (IACUC) of National Taiwan Sport University (Approval No. IACUC-10823) and were conducted in accordance with national regulations and institutional guidelines on animal care and use. A total of forty male C57BL/6J mice were used, including eight young mice (8 weeks old) and thirty-two aging mice (85 weeks old), all obtained from BioLASCO (Charles River Licensee Corp., Yi-Lan, Taiwan). The mice were randomly divided into five groups: a young sedentary control group (YS), an aging sedentary control group (AS), an aging group receiving aerobic exercise training (AE), an aged group receiving NMN supplementation at 300 mg/kg/day (ASNMN), and an aged group receiving both aerobic exercise and NMN supplementation (AENMN). Throughout the six-week intervention period, all animals were housed in standard cages under controlled environmental conditions (22 ± 2 °C, 60–70% humidity, 12-h light/dark cycle), with free access to distilled water and a standard laboratory chow diet (No. 5001; PMI Nutrition International, Brentwood, MO, USA; containing approximately 23% crude protein, 4.5% crude fat, 6% crude fiber, and 44% carbohydrate). Water and food intake were measured weekly by weighing bottles and chow, correcting for spillage, and values were converted to average daily intake per mouse across the 6-week intervention. After completion of the 6-week intervention, body composition was assessed on Day 43, followed by functional performance assessments on Days 44–49, including the treadmill endurance test, forelimb grip strength, weights test, Kondziela’s inverted screen test, functional strength test, and rotarod balance test (Figure 1). These tests were performed over a 6-day period, with at least 24 h of rest between major tests. Subsequently, mice were subjected to an oral glucose tolerance test (OGTT) on Day 51 and indirect calorimetry for energy expenditure and spontaneous activity on Days 53–55. Finally, animals were euthanized on Day 56 for blood collection, biochemical analyses, tissue harvesting, histological staining, and molecular assays.

### 2.2. NMN Administration

β-Nicotinamide mononucleotide (NMN; California Gold Nutrition, Irvine, CA, USA) was freshly prepared in autoclaved water immediately before use. The solution was administered once daily by oral gavage at a dosage of 300 mg/kg body weight throughout the 6-week intervention period. This dosage was selected based on previously published evidence demonstrating its effectiveness in aged mice [26].

### 2.3. Exercise Intervention Protocol

During the 6-week intervention period, all mice in the exercise groups underwent aerobic exercise training using a motor-driven treadmill (model MK-680; Muromachi Kikai, Tokyo, Japan). The protocol was adapted from a previously established scheme [29]. An initial 2-day acclimation phase was implemented, during which the mice ran for 10 min at a 5° gradient and a speed of 10 m/min. Thereafter, the treadmill gradient was increased to 10°, and both running speed and running duration were progressively adjusted on a weekly basis, reaching 20 m/min and 20 min by the end of week 6. To encourage compliance, an electrified grid was positioned at the rear of each treadmill lane, delivering a mild stimulus (0.2 mA, 200 ms pulses, 1 Hz) to stationary mice.

### 2.4. Body Composition

Body composition, including fat mass, lean mass, and free body fluid, was assessed once on Day 43, immediately following the completion of the 6-week intervention (Day 42), using a time-domain nuclear magnetic resonance (TD-NMR) system (Minispec LF50, Bruker, Bremen, Germany). Prior to measurement, the device was calibrated following the manufacturer’s guidelines. Each mouse was gently placed into a 50 mm diameter plastic cylinder and restrained using a plunger to minimize movement. The cylinder was then inserted into the measurement chamber, and data collection was completed within approximately two minutes. Composition values were derived from radiofrequency signals generated by hydrogen nuclei in soft tissues, based on differences in relaxation times and signal amplitudes.

### 2.5. Endurance Exercise Performance Test 

The treadmill endurance test was performed once on Day 44, following the completion of the 6-week intervention (Day 42), to assess aerobic capacity. To maintain consistent motivation during the test, a mild electrical stimulus was applied using a shock grid under veterinary supervision. Mice that were not part of the exercise training groups underwent pre-test acclimation by running on the treadmill for 5 min per day at a speed of 10 m/min with a 5° incline for one week prior to the test.

During the endurance test, the treadmill was set at a 5° gradient with an initial speed of 15 m/min. The speed was increased incrementally by 2 m/min every 4 min until the animal could no longer maintain running and remained on the shock grid for 5 s. This point was defined as the time to exhaustion [30].

### 2.6. Forelimb Grip Strength

Forelimb grip strength was measured once on Day 45, following the completion of the 6-week intervention (Day 42), using a low-force testing system (Model RX-5, Aikoh Engineering, Nagoya, Japan) equipped with a horizontal pull rod measuring 2 mm in diameter and 7.5 cm in length. Each mouse was gently held by the base of the tail and guided toward the rod to allow its forepaws to grasp it. Once the mouse grasped the rod, it was gently pulled backward in the opposite direction to elicit maximal resistance. The peak force exerted was recorded across 10 consecutive trials, and the highest value was used for subsequent analysis [31].

### 2.7. Weights Test

The weights test was performed once on Day 46, following the completion of the 6-week intervention (Day 42), using a modified method previously described to assess muscle strength via voluntary forelimb grasping of weighted steel wool balls [32]. A base steel wool ball weighing 15 g was constructed to facilitate gripping, and additional 15 g segments were sequentially added to increase resistance. During testing, each mouse was gently held at approximately one-third of its tail length, allowing it to naturally grasp the steel wool ball.

Once the mouse successfully lifted the ball off the surface, the timing commenced. The upper limit for each lifting attempt was set at 3 s. Three attempts constituted one test stage. Upon successful completion of a stage, the mouse was given a 10-s rest period before an additional 15 g segment was added to the load. If the mouse failed to lift the weight within 3 s in all three attempts at a given stage, the test was concluded. The final grip score was calculated as the product of the number of completed stages and the number of successful repetitions, in accordance with the original protocol [32].

### 2.8. Kondziela’s Inverted Screen Test

The inverted screen test was performed once on Day 47, following the completion of the 6-week intervention (Day 42), to evaluate neuromuscular strength and endurance in aging mice, following the method described by Kondziela [32]. The test apparatus consisted of a 43 × 43 cm wire mesh screen, constructed with 12 mm square grids using 1 mm diameter stainless steel wires. Each mouse was gently placed at the center of the screen, which was then slowly and evenly inverted to a 180° angle and held approximately 40 to 50 cm above a padded surface. The latency to fall was recorded, with a maximum cutoff time of 60 s. Performance was scored based on fall latency using the following standardized scale: a fall within 1 to 10 s was assigned 1 point, 11 to 25 s was assigned 2 points, 26 to 60 s was assigned 3 points, and sustained suspension beyond 60 s was assigned 4 points [32].

### 2.9. Functional Strength Test

The functional strength test was performed once on Day 48, following the completion of the 6-week intervention (Day 42), using a previously validated method to evaluate isometric muscular endurance in aging mice [33]. The apparatus and mesh were identical to those used in training, providing consistency and minimizing variability. Lead sheets equivalent to 10% of each mouse’s body weight were placed on the dorsal surface. Timing began once the load was applied and ended when the mouse released both hind limbs from the mesh. This test provided an indirect but robust index of isometric strength capacity in a context mimicking resistance loading in aging animals.

### 2.10. Rotarod Balance Performance Test

The rotarod balance performance test was performed once on Day 49, after completion of the 6-week intervention (Day 42). All mice were subjected to an adaptive session using a roller-type treadmill apparatus to familiarize them with the equipment prior to the training protocol. After a week of learning and adaptation, the test was performed at an initial speed of 4 rpm and accelerated to 40 rpm within 5 min, and the time from the start of the axis rotation to the mouse falling was recorded [34].

### 2.11. Oral Glucose Tolerance Test (OGTT)

The oral glucose tolerance test (OGTT) was performed once on Day 51, after completion of the 6-week intervention (Day 42). All mice were subjected to oral glucose tolerance testing following a 12 h overnight fast. Blood samples were collected from the tail vein, and glucose levels were immediately measured using the ONETOUCH^®^ Ultra Plus Flex™ glucometer (LifeScan Inc., Milpitas, CA, USA). After baseline glucose measurement, each mouse received an oral gavage of 20% glucose solution (2 g/kg body weight). Subsequent blood glucose concentrations were measured at 15, 30, 60, and 120 min post-administration to evaluate systemic glucose clearance and tolerance.

### 2.12. Measurement of Metabolic Rate

Whole-body metabolic rate and associated physiological parameters were evaluated once on Days 53–55, after completion of the 6-week intervention (Day 42), using the Oxymax indirect calorimetry system (Columbus Instruments, Columbus, OH, USA) at the Taiwan Mouse Clinic. Mice were individually housed in metabolic cages and acclimated for 24 h prior to data collection. Following this acclimation period, all animals remained under controlled environmental conditions with unrestricted access to food and water. Continuous measurements were recorded over a 48-h period to capture steady-state metabolic activity. The primary variables analyzed included oxygen consumption (VO_2_), carbon dioxide production (VCO_2_), respiratory exchange ratio (RER), and spontaneous locomotor activity, which was measured using wheel-running activity integrated into the calorimetry system.

### 2.13. Tissue and Sample Collection

On Day 56, after completion of the 6-week intervention (Day 42), all animals were fasted for 12 h and euthanized with 95% CO_2_. Blood was collected via cardiac puncture for biochemical analyses. Then, major organs including the liver, kidneys, heart, lungs, brain, brown adipose tissue (BAT), epididymal fat pad (EFP), as well as skeletal muscles (gastrocnemius and soleus), were carefully excised, weighed, and either fixed in formalin for histological analysis or snap-frozen for molecular assays.

### 2.14. Measurement of Biochemical Parameters

Serum samples obtained at sacrifice (Day 56) were centrifuged at 1500× *g* for 15 min at 4 °C to separate serum. Serum aspartate transferase (AST), alanine transferase (ALT), creatine phosphokinase (CPK), total protein (TP), albumin (ALB), blood urea nitrogen (BUN), total cholesterol (TC), triglyceride (TG), high-density lipoprotein cholesterol (HDL-C), and low-density lipoprotein cholesterol (LDL-C) levels were analyzed using commercial enzymatic kits (DiaSys Diagnostic Systems GmbH, Holzheim, Germany) and quantified with a Hitachi Biochemical Analyzer 7150 (Hitachi Chemical Co., Ltd., Tokyo, Japan). Nicotinamide phosphoribosyltransferase (NAMPT) concentrations were determined using a commercial ELISA kit (NAMPT [visfatin/PBEF] Mouse/Rat Dual ELISA Kit, catalog AG-45A-0007YEK-KI01; AdipoGen Life Sciences, Füllinsdorf, Switzerland), according to the manufacturer’s instructions.

### 2.15. Western Blotting

Skeletal muscle samples from the hind limb (gastrocnemius and soleus) collected at sacrifice (Day 56) were homogenized in T-PER tissue protein extraction reagent (Thermo Fisher Scientific, Waltham, MA, USA), supplemented with a protease inhibitor cocktail and PhosSTOP phosphatase inhibitors (Roche, Basel, Switzerland). Homogenates were centrifuged at 12,500 rpm for 25 min at 4 °C to obtain clarified lysates. Protein concentrations were quantified using the Bradford protein assay (Bio-Rad Laboratories, Hercules, CA, USA). For Western blotting, 30 µg of total protein from each sample was separated on 10% SDS-PAGE gels and transferred to nitrocellulose membranes. Membranes were probed with specific primary antibodies, and GAPDH was used as the internal loading control. Protein bands were visualized using enhanced chemiluminescence (ECL; Amersham Biosciences, Amersham, UK), and molecular weight markers were included. Densitometric analysis was performed using the ImageJ software (version 1.53k; NIH, Bethesda, MD, USA), and target protein expression was normalized to GAPDH. The primary antibody used in this study was anti-SirT1 (C14H4) (1:500; #2496; Cell Signaling Technology, Danvers, MA, USA). Protein expression was quantified by densitometric analysis using the ImageJ software (National Institutes of Health, Bethesda, MD, USA).

### 2.16. Statistical Analysis

All data are expressed as mean ± standard deviation (SD). Statistical analyses were performed using the SAS software (version 9.0; SAS Institute Inc., Cary, NC, USA). Group differences were analyzed by one-way analysis of variance (ANOVA), followed by Duncan’s multiple range test for post hoc comparisons. Statistical significance was set at *p* < 0.05.

## 3. Results

### 3.1. General Characteristics of Aging Mice with NMN Supplementation and Exercise Training

Throughout the experimental period, no significant differences in average body weight were noted among the aging groups; however, all aging groups maintained significantly higher body weights compared to the young control group (*p* < 0.05) (Figure 2). We observed that the water and food intake of the aging group was significantly higher than that of young mice. Among them, the mice in the NMN group that were supplemented during the experiment had significantly lower food intake and calories than the mice that were not supplemented with NMN (*p* < 0.05) (Table 1).

Body weight presented in Table 1 reflects the final measurement taken after a 12-h fasting period immediately prior to euthanasia. Aged mice exhibited significantly higher final body weight compared to young controls (*p* < 0.05). In addition, the weights of major organs and adipose tissue, including the heart, liver, kidneys, brain, and epididymal fat pad (EFP), were significantly greater in all aged groups than in the young group (Table 1). Among these, EFP weight was significantly lower in aging mice that underwent exercise training compared to sedentary aging mice. The weights of the lung, muscle, and BAT did not significantly differ among all groups. Because tissue weight may be influenced by overall body weight, we divided tissue weight by body weight to calculate the relative percentage of body weight and found similar results to absolute tissue weight.

### 3.2. Effect of NMN Supplementation and Exercise Training on the Body Composition of Aging Mice

To assess whole-body composition, mice were analyzed using a time-domain nuclear magnetic resonance (TD-NMR) system prior to sacrifice. As shown in Table 2, body weight, fat mass, lean mass, and free body fluid were significantly increased in all aging groups compared to the young controls (*p* < 0.05). However, statistically significant differences in these parameters were not observed among the groups. These results indicate that neither NMN supplementation nor exercise training alone or in combination significantly altered overall body composition in aged mice.

### 3.3. Effect of NMN Supplementation and Exercise Training on Energy Metabolism and Physical Activity

VO_2_ levels during the light, dark, and 24-h periods are presented in Figure 3A. During the light phase, VO_2_ levels were significantly higher in the AE (*p* = 0.0127), ASNMN (*p* = 0.0478), and AENMN (*p* < 0.0001) groups compared to the AS group, with the AENMN group showing greater VO_2_ levels than both the AE (*p* = 0.0337) and ASNMN groups (*p* = 0.0084). In the dark phase, VO_2_ levels were significantly lower in AS compared to YS (*p* < 0.0001), whereas the levels in the AENMN group remained significantly higher than those in the AS (*p* < 0.0001), AE (*p* = 0.0398), and ASNMN groups (*p* = 0.0104). Over the entire 24-h period, VO_2_ was reduced in the AS group relative to the YS group (*p* < 0.0001), whereas both the AE (*p* = 0.0167) and AENMN groups (*p* < 0.0001) exhibited significantly improved VO_2_ levels. Notably, the AENMN group exhibited higher VO_2_ levels than AE (*p* = 0.048) and ASNMN (*p* = 0.0105) groups; however, these levels did not differ significantly from those in the YS group (*p* = 0.1424).

VCO_2_ levels during the light, dark, and 24-h periods are presented in Figure 3B. During the light phase, VCO_2_ was significantly higher in the AE (*p* = 0.0358) and AENMN (*p* = 0.0074) groups compared to the AS group. In the dark phase, the AS group exhibited significantly lower VCO_2_ values than the YS group (*p* = 0.003), and the levels in the ASNMN group were also significantly lower than those in the YS group (*p* = 0.0002). Across the 24-h period, VCO_2_ levels were significantly reduced in the AS group compared to the YS group (*p* = 0.0046), whereas no significant differences were observed among the aging intervention groups.

The values of the RER during the light, dark, and 24-h periods are presented in Figure 3C. During the light phase, the RER was significantly higher in the AS (*p* < 0.0001) and AE (*p* = 0.0007) groups compared to the YS group, whereas the AENMN group showed a significantly lower RER than the AS (*p* = 0.001), AE (*p* = 0.0169), and ASNMN (*p* = 0.0025) groups. In the dark phase, the AS group exhibited a significantly elevated RER compared to YS (*p* < 0.0001), whereas the AENMN group had a significantly lower RER than the AS (*p* < 0.0001), AE (*p* = 0.0006), and ASNMN (*p* = 0.0006) groups. Over the 24-h period, the AS group had a significantly higher RER than the YS (*p* < 0.0001) and AE (*p* = 0.0455) groups. The AENMN group showed a significantly lower RER than the AS (*p* < 0.0001) and AE (*p* = 0.0005) groups, and its RER was not significantly different from that noted for the YS group (*p* = 0.8251).

EE values during the light, dark, and 24-h periods are presented in Figure 3D. During the light phase, the EE was significantly lower in the AS (*p* < 0.0001), AE (*p* = 0.0002), and ASNMN (*p* = 0.0009) groups compared to the YS group. The AENMN group had significantly higher EE values than the AS (*p* = 0.0061), AE (*p* = 0.0136), and ASNMN (*p* = 0.0351) groups. In the dark phase, EE values remained significantly reduced in the AS, AE, and ASNMN groups relative to the YS group (*p* < 0.0001), whereas the AENMN group showed a significantly higher EE than the AS (*p* = 0.0498), AE (*p* = 0.0014), and ASNMN (*p* < 0.0001) groups. Across the 24-h period, EE values were significantly lower in the AS, AE, and ASNMN groups compared to the YS group (*p* < 0.0001), and the AENMN group showed a significantly higher EE level than the AS (*p* = 0.0009), AE (*p* = 0.0093), and ASNMN (*p* = 0.0027) groups.

Data on wheel-running activity and usage time are presented in Figure 4A,B. Spontaneous locomotor activity, the AS group showed a significant 87.8% reduction compared to the YS group (*p* < 0.0001). The AE (*p* < 0.0001), ASNMN (*p* = 0.0077), and AENMN (*p* < 0.0001) groups all displayed significantly improved activity compared to the AS group. Notably, the AENMN group showed the greatest recovery, with a 2.85-fold increase over the AS group (*p* = 0.0005). Similarly, the average wheel usage time per hour was significantly reduced in the AS group compared to the YS group (*p* < 0.0001). All interventions increased usage time relative to the AS group, with the AE, ASNMN, and AENMN groups showing 1.73-fold (*p* < 0.0001), 1.37-fold (*p* = 0.0069), and 1.77-fold (*p* < 0.0001) increases, respectively. The AENMN group showed the greatest improvement, with a 77.5% increase over the AS group (*p* = 0.0038).

### 3.4. Effect of NMN Supplementation and Exercise Training on Oral Glucose Tolerance Test Results

At baseline, no significant differences in blood glucose concentrations were observed among groups. Following glucose administration after the 6-week intervention, both the ASNMN and AENMN groups exhibited significantly lower glucose levels over time compared to the AS group. The AENMN group demonstrated improved glucose tolerance, as reflected by a 17.48%, 17.50%, and 14.95% reduction in AUC values compared to the YS (*p* = 0.0134), AS (*p* = 0.0133), and AE (*p* = 0.0376) groups, respectively (Table 3).

### 3.5. Effects of NMN Supplementation and Exercise Training on Endurance Performance in Aging Mice

After completion of the 6-week intervention, the treadmill endurance test was performed on Day 44 to assess aerobic capacity. The running times for the YS, AS, AE, ASNMN, and AENMN groups were 12.37± 1.65, 7.14 ± 2.49, 12.66 ± 1.30, 9.50 ± 2.18, and 12.44 ± 2.99 min, respectively. Compared to the YS group, the running time was significantly lower in the AS and ASNMN groups by 42.32% (*p* < 0.0001) and 23.19% (*p* = 0.0392), respectively. In contrast, the AE and AENMN groups showed significantly greater endurance than the AS group, with 1.77-fold (*p* < 0.0001) and 1.74-fold (*p* < 0.0001) increases, respectively (Figure 5).

### 3.6. Effects of NMN Supplementation and Exercise Training on Muscle Strength and Physical Function in Aging Mice

In the grip strength test (Figure 6A), grip strength was significantly lower in the AS and ASNMN groups (*p* = 0.0091 and *p* = 0.0338, respectively) compared to the YS group. Additionally, no significant differences were found among the aging groups. Regarding relative grip strength (%) (Figure 6B), the YS group showed significantly higher values than all aging groups (*p* < 0.0001), with no significant differences noted among the aged groups.

The weights test values for the YS, AS, AE, ASNMN, and AENMN groups were 8.38 ± 2.72, 5.75 ± 3.24, 9.25 ± 2.31, 8.88 ± 2.36, and 8.75 ± 1.75, respectively (score). Compared with the AS group, the values of the YS, AE, ASNMN, and AENMN groups were significantly increased by 1.46-fold (*p* = 0.0450) 1.61-fold (*p* = 0.0089), 1.54-fold (*p* = 0.0183) and 1.52-fold (*p* = 0.0231), respectively (Figure 7A).

In the Kondziela’s inverted screen test (Figure 7B), the YS group showed significantly higher scores than the AS (*p* = 0.0006), AE (*p* = 0.0323), ASNMN (*p* = 0.0113), and AENMN groups (*p* = 0.0113), with no significant differences observed among the aged groups (*p* > 0.05).

In the functional strength test (Figure 7C), the AS group exhibited significantly shorter maximum holding time compared to the YS (*p* < 0.0001), AE (*p* = 0.0002), ASNMN (*p* = 0.0002), and AENMN (*p* < 0.0001) groups. In contrast, the AENMN group showed the highest performance, with significantly longer holding time than the YS (*p* = 0.0002), AE (*p* = 0.0086), and ASNMN (*p* < 0.0001) groups. No significant differences were observed among the YS, AE, and ASNMN groups.

In the rotarod balance test (Figure 7D), the AS and ASNMN groups exhibited reduced balance time compared to the other groups; however, the differences were not statistically significant.

### 3.7. Effect of NMN Supplementation and Exercise Training on Biochemical Variables at the End of the Experiment

At the end of the experiment, no significant differences in AST, ALT, TP, ALB, TC, HDL, LDL, and CK levels were observed among groups (Table 4). However, all the aging mice had a significantly higher BUN concentration than YS group (*p* < 0.05). Both NMN and exercise training intervention significantly reduced blood TG concentrations compared with the YS and AS groups (*p* < 0.05). The YS group had significantly higher NAMPT levels than all other groups (*p* < 0.05), and the AE, ASNMN, and AENMN groups had significantly increased NAMPT levels compared with AS group, with values increased by 1.61-fold (*p* < 0.0001), 1.63-fold (*p* < 0.0001), and 1.74-fold (*p* < 0.0001), respectively.

### 3.8. Effect of NMN Supplementation and Exercise Training on Western Blot Analysis of Gastrocnemius Muscle

As shown in Figure 8, the relative expression of SirT1 protein in the gastrocnemius muscle, normalized to GAPDH, was 1.00 ± 0.00 (YS), 1.60 ± 0.13 (AS), 1.51 ± 0.39 (AE), 1.95 ± 0.17 (ASNMN), and 1.94 ± 0.63 (AENMN). SirT1 expression in the YS group was significantly lower than in the ASNMN and AENMN groups (*p* = 0.0408). However, no statistically significant differences were observed among the aging groups (AS, AE, ASNMN, and AENMN), indicating that the combined treatment did not yield additive effects.

## 4. Discussion

Aerobic exercise has been well documented to enhance cardiovascular function and mitigate declines in strength, mobility, balance, and endurance commonly observed in older adults [35]. In parallel, interest in the geroprotective potential of NMN has grown considerably in recent years [36]. However, studies investigating whether NMN supplementation combined with aerobic exercise produces synergistic effects remain limited. In the current study, we administered NMN supplementation to aging mice for six consecutive weeks combined with aerobic exercise training. The combined intervention was associated with improvements in aerobic endurance and glucose tolerance, along with modest benefits for muscular strength. Increases in skeletal muscle SirT1 expression were also detected in NMN-supplemented groups.

Beyond improvements in physical performance, the combined intervention (AENMN) significantly enhanced systemic energy metabolism. This was demonstrated by increased VO_2_, elevated energy expenditure, and greater locomotor activity. These changes were accompanied by a reduced respiratory quotient, indicating a greater reliance on lipid oxidation as the predominant energy source during metabolic assessment, and suggesting improved mitochondrial efficiency. This interpretation aligns with previous reports indicating that NMN supplementation supports mitochondrial integrity and oxidative function, whereas aerobic exercise facilitates NAD^+^ biosynthesis and improves metabolic flexibility [15,26,37]. Together, these outcomes support the presence of a synergistic effect between NMN and exercise in restoring metabolic resilience during aging.

To date, few studies have comprehensively evaluated the effects of NMN on skeletal muscle mass or neuromuscular performance in aging models. In this study, we examined muscle strength, endurance, and aerobic capacity to assess these outcomes. Although some improvements in muscle strength and endurance were observed in specific intervention groups, the effects were limited and did not consistently reach statistical significance across all tests, suggesting that NMN supplementation and moderate aerobic exercise alone may have only modest benefits on neuromuscular performance in aging mice. It is also possible that the six-week intervention period was too short to capture slower or structural skeletal muscle adaptations, which may partly explain the limited strength improvements observed in this study. Likewise, the absence of significant changes in body composition across groups may also reflect the relatively short 6-week duration, which may not have been sufficient to induce measurable alterations in fat or lean mass. Aerobic exercise alone has minimal effects on muscle strength unless training intensity or type is sufficiently progressive; however, it remains a proven strategy to enhance aerobic capacity, mitochondrial ATP production, and cardiovascular function [38]. Consistent with these findings, our data showed marked improvements in aerobic endurance among mice undergoing treadmill-based exercise training, irrespective of whether NMN supplementation was included.

Beyond improvements in aerobic performance, NMN enhances mitochondrial function, vascular integrity, and capillary density in rodent models [39]. Skeletal muscle is particularly vulnerable to age-related vascular decline, including reduced neovascularization and capillary rarefaction, both of which contribute to loss of muscle mass and endurance [40]. Age-related declines in NAD^+^ may impair vascular function and endurance, whereas NMN—as a precursor of NAD^+^—can restore NAD^+^ availability and activate SirT1, a regulator of vascular health and angiogenesis [41]. SirT1 in endothelial cells modulates pro-angiogenic signaling, promotes capillary density and blood flow, and may thereby improve oxygen delivery and endurance [42]. Although in our study SirT1 expression did not differ significantly among aged intervention groups, both ASNMN and AENMN groups exhibited higher levels compared with YS. These results suggest that NMN may contribute to SirT1 upregulation, though the combined effect with exercise was not statistically significant. SirT1 is also known to promote mitochondrial biogenesis, reduce inflammation, and regulate metabolic homeostasis [43,44,45,46,47,48], and its upregulation has been interpreted as a compensatory response to oxidative stress [49,50]. These mechanisms are consistent with prior reports that NMN restores NAD^+^ availability, activates SirT1 and AMPK signaling, and supports systemic metabolic resilience in aged or metabolically impaired models [26,51].

We also observed significantly elevated blood NAMPT levels following NMN supplementation. NAMPT, the rate-limiting enzyme in the NAD^+^ salvage pathway, converts nicotinamide to NMN and thereby enhances NAD^+^ biosynthesis [40]. This upregulation may promote mitochondrial biogenesis via PGC-1α and enhance oxidative metabolism and fatty acid utilization [52]. Through regulation of lipid-related genes such as PPARs, SirT1 may also contribute to reduced fat accumulation [53]. Consistent with this, reductions in accessory testicular fat were observed in NMN-treated mice; however, systemic changes in body composition were not significant. Although these results and prior studies [26,39,40,41,42,43,44,45,46,47,48,49,50,51,52,53] suggest potential vascular and metabolic benefits of NMN, it is important to note that we did not directly assess angiogenesis, capillary density, mitochondrial biogenesis, or glucose metabolism markers in this study. Therefore, these mechanistic interpretations remain speculative and require confirmation in future investigations.

NMN has primarily been studied for its effects on aging, mitochondrial integrity, cardiovascular health, DNA repair, and cognitive function [36]. In the present study, six weeks of NMN supplementation combined with aerobic exercise was associated with improvements in aerobic endurance, glucose tolerance, and fat oxidation in aging mice. These outcomes, which include enhanced aerobic capacity, better glucose regulation, and favorable shifts in fat metabolism, are all relevant to mitigating age-related metabolic and functional decline. Nevertheless, improvements in muscle strength were not consistently observed, underscoring the need for longer interventions and larger cohorts to clarify the effects of NMN and exercise on neuromuscular outcomes.

## 5. Conclusions

In conclusion, this study provides evidence that six weeks of NMN supplementation combined with aerobic exercise improved aerobic endurance, glucose tolerance, and systemic energy metabolism in naturally aged mice. These adaptations were accompanied by increased oxygen consumption, elevated energy expenditure, upregulated NAMPT and SirT1 expression, and reduced respiratory exchange ratio, suggesting enhanced mitochondrial efficiency. However, the improvements in muscle strength and endurance were modest, and no significant changes in overall body composition were detected. These outcomes potentially reflect the relatively short intervention period and limited sample size, which could constrain the ability to detect more subtle or long-term adaptations. Collectively, our findings suggest that NMN, particularly when combined with aerobic exercise, may help to sustain metabolic health and delay age-related functional decline.

## Figures and Tables

**Figure 1 nutrients-17-03148-f001:**
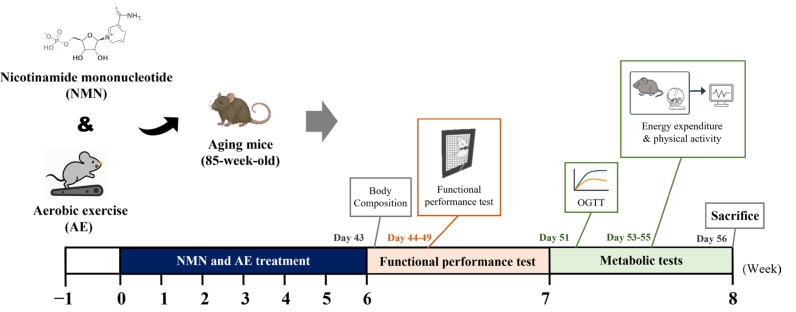
Experimental design. Mice underwent a six-week intervention of NMN supplementation and/or aerobic exercise (Days 0–42). After completion, body composition was assessed (Day 43), followed sequentially by functional performance tests (Days 44–49; treadmill endurance, forelimb grip strength, weights, Kondziela’s inverted screen, functional strength, and rotarod balance), an oral glucose tolerance test (OGTT, Day 51), indirect calorimetry for energy expenditure and physical activity (Days 53–55), and final sacrifice for blood and tissue analyses (Day 56).

**Figure 2 nutrients-17-03148-f002:**
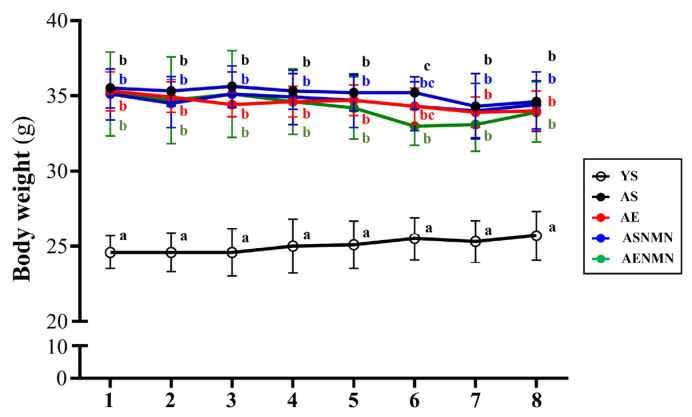
Effect of NMN supplementation and exercise training on body weight. Data are the mean ± SD for n = 8 mice in each group. Different superscript letters (a, b, and c) indicate statistically significant differences among groups (*p* < 0.05, one-way ANOVA with Duncan’s post hoc test). Groups sharing the same letter are not significantly different. YS, young sedentary; AS, aged sedentary; AE, aged with exercise; ASNMN, aged with NMN supplementation; AENMN, aged with combined NMN and exercise.

**Figure 3 nutrients-17-03148-f003:**
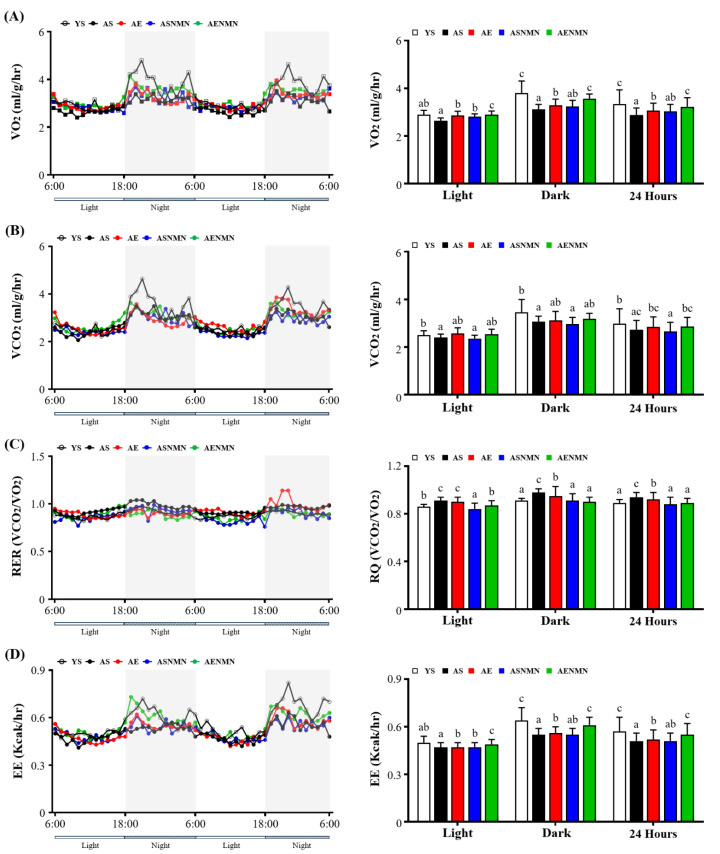
Effect of NMN supplementation and exercise training on (**A**) oxygen consumption (VO_2_), (**B**) carbon dioxide production (VCO_2_), (**C**) respiratory exchange ratio (RER), and (**D**) energy expenditure (EE). Data are expressed as mean ± SD for n = 4 mice in each group. Different superscript letters (a, b, and c) indicate statistically significant differences among groups (*p* < 0.05, one-way ANOVA with Duncan’s post hoc test). Groups sharing the same letter are not significantly different. YS, young sedentary; AS, aged sedentary; AE, aged with exercise; ASNMN, aged with NMN supplementation; AENMN, aged with combined NMN and exercise.

**Figure 4 nutrients-17-03148-f004:**
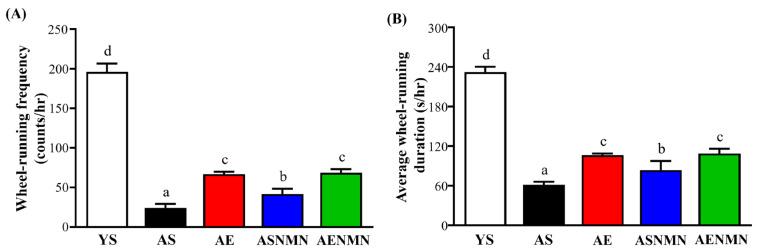
Effects of NMN supplementation and aerobic exercise training on spontaneous physical activity: (**A**) frequency of wheel-running activity per hour, and (**B**) average wheel usage time per hour. Data are expressed as mean ± SD for n = 4 mice in each group. Different superscript letters (a, b, c and d) indicate statistically significant differences between groups (*p* < 0.05, one-way ANOVA with Duncan’s post hoc test). Groups sharing the same letter are not significantly different. YS, young sedentary; AS, aged sedentary; AE, aged with exercise; ASNMN, aged with NMN supplementation; AENMN, aged with combined NMN and exercise.

**Figure 5 nutrients-17-03148-f005:**
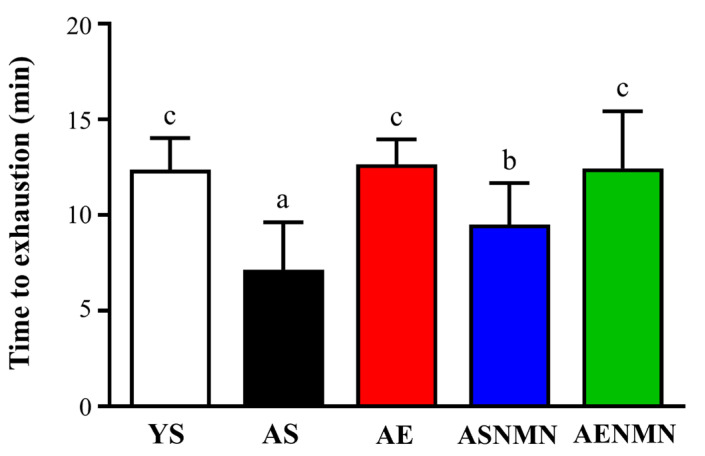
Effects of NMN supplementation and aerobic exercise training on treadmill endurance test results. Data are expressed as mean ± SD for n = 8 mice in each group. Different superscript letters (a, b, and c) indicate statistically significant differences among groups (*p* < 0.05, one-way ANOVA with Duncan’s post hoc test). Groups sharing the same letter are not significantly different. YS, young sedentary; AS, aged sedentary; AE, aged with exercise; ASNMN, aged with NMN supplementation; AENMN, aged with combined NMN and exercise.

**Figure 6 nutrients-17-03148-f006:**
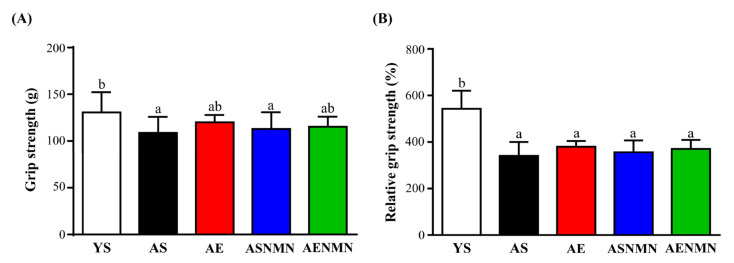
Effects of NMN supplementation and aerobic exercise training on (**A**) forelimb grip strength and (**B**) relative forelimb grip strength (%). Data are expressed as mean ± SD for n = 8 mice in each group. Different superscript letters (a and b) indicate statistically significant differences among groups (*p* < 0.05, one-way ANOVA with Duncan’s post hoc test). Groups sharing the same letter are not significantly different. Relative grip strength = (grip strength/body weight) × 100%. YS, young sedentary; AS, aged sedentary; AE, aged with exercise; ASNMN, aged with NMN supplementation; AENMN, aged with combined NMN and exercise.

**Figure 7 nutrients-17-03148-f007:**
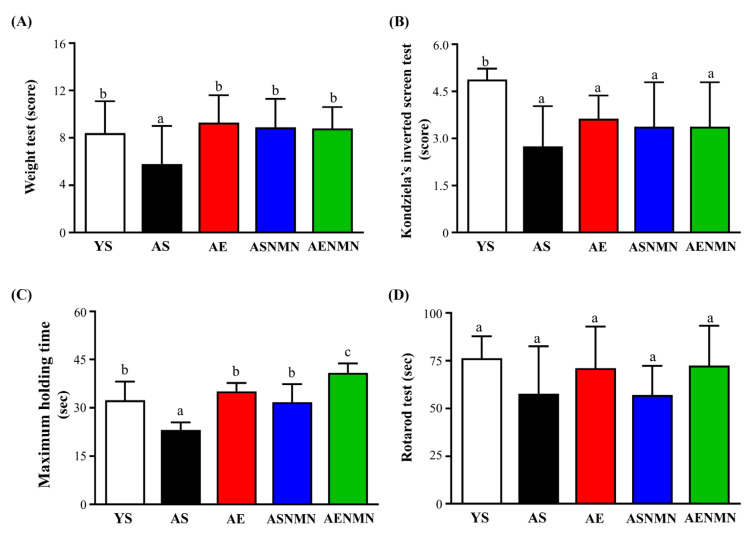
Effects of NMN supplementation and aerobic exercise training on (**A**) weights test, (**B**) Kondziela’s inverted screen test, (**C**) functional strength test, and (**D**) rotarod balance test. Data are expressed as mean ± SD for n = 8 mice in each group. Different superscript letters (a, b, and c) indicate statistically significant differences among groups (*p* < 0.05, one-way ANOVA with Duncan’s post hoc test). Groups sharing the same letter are not significantly different. YS, young sedentary; AS, aged sedentary; AE, aged with exercise; ASNMN, aged with NMN supplementation; AENMN, aged with combined NMN and exercise.

**Figure 8 nutrients-17-03148-f008:**
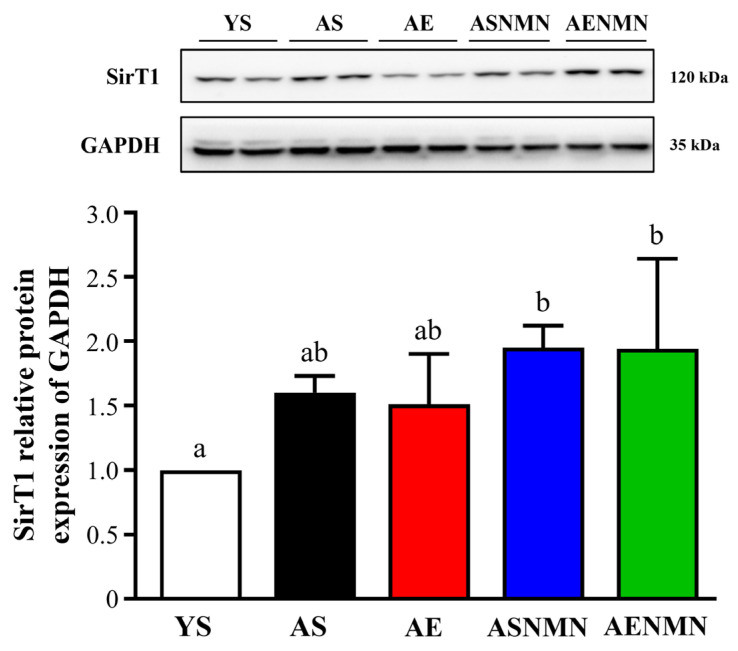
Effects of NMN supplementation and aerobic exercise training on SirT1 protein expression in skeletal muscle, as determined using Western blot analysis. Data are expressed as mean ± SD for n = 2 mice in each group. Different superscript letters (a and b) indicate statistically significant differences between groups (*p* < 0.05, one-way ANOVA with Duncan’s post hoc test). Groups sharing the same letter are not significantly different. YS, young sedentary; AS, aged sedentary; AE, aged with exercise; ASNMN, aged with NMN supplementation; AENMN, aged with combined NMN and exercise; SirT1, sirtuin 1.

**Table 1 nutrients-17-03148-t001:** Effect of NMN supplementation and exercise training on various parameters in aging mice.

Characteristic	YS	AS	AE	ASNMN	AENMN
Final BW (g)	24.03 ± 1.77 ^a^	32.16 ± 1.98 ^b^	31.63 ± 1.06 ^b^	31.71 ± 1.72 ^b^	31.21 ± 1.68 ^b^
Water intake (mL/mouse/day)	4.51 ± 0.92 ^a^	8.26 ± 1.32 ^d^	7.17 ± 0.83 ^b^	7.69 ± 1.16 ^c^	7.33 ± 1.02 ^bc^
Food intake (g/mouse/day)	4.16 ± 0.62 ^a^	5.68 ± 0.79 ^d^	5.78 ± 0.81 ^d^	4.85 ± 0.89 ^b^	5.32 ± 0.82 ^c^
Calories intake (kcal/mouse/day)	16.44 ± 2.46 ^a^	22.43 ± 3.1 ^c^	22.85 ± 3.19 ^c^	19.16 ± 3.53 ^b^	18.62 ± 2.89 ^b^
Absolute weight					
Liver (g)	1.04 ± 0.14 ^a^	1.61 ± 0.16 ^b^	1.52 ± 0.17 ^b^	1.48 ± 0.15 ^b^	1.68 ± 0.39 ^b^
Kidney (g)	0.31 ± 0.02 ^a^	0.46 ± 0.03 ^b^	0.48 ± 0.02 ^b^	0.46 ± 0.04 ^b^	0.46 ± 0.03 ^b^
Muscle (g)	0.29 ± 0.02 ^a^	0.30 ± 0.02 ^a^	0.30 ± 0.02 ^a^	0.30 ± 0.02 ^a^	0.30 ± 0.02 ^a^
EFP (g)	0.36 ± 0.06 ^a^	0.64 ± 0.16 ^c^	0.51 ± 0.12 ^b^	0.64 ± 0.06 ^c^	0.49 ± 0.14 ^b^
Heart (g)	0.13 ± 0.03 ^a^	0.19 ± 0.02 ^b^	0.19 ± 0.02 ^b^	0.21 ± 0.04 ^b^	0.20 ± 0.01 ^b^
Lung (g)	0.18 ± 0.04 ^a^	0.20 ± 0.03 ^b^	0.20 ± 0.02 ^b^	0.21 ± 0.04 ^b^	0.20 ± 0.05 ^b^
Brain (g)	0.46 ± 0.02 ^a^	0.49 ± 0.01 ^b^	0.49 ± 0.02 ^b^	0.48 ± 0.03 ^b^	0.49 ± 0.02 ^b^
BAT (g)	0.06 ± 0.01 ^ab^	0.05 ± 0.01 ^ab^	0.05 ± 0.01 ^a^	0.05 ± 0.01 ^ab^	0.06 ± 0.02 ^b^
Relative weight					
Liver (%)	3.96 ± 0.53 ^a^	6.13 ± 0.59 ^b^	5.79 ± 0.66 ^b^	5.65 ± 0.58 ^b^	6.39 ± 1.48 ^b^
Kidney (%)	1.19 ± 0.30 ^a^	1.77 ± 0.10 ^b^	1.84 ± 0.09 ^b^	1.77 ± 0.14 ^b^	1.77 ± 0.13 ^b^
Muscle (%)	1.09 ± 0.09 ^a^	1.15 ± 0.10 ^a^	1.15 ± 0.07 ^a^	1.15 ± 0.09 ^a^	1.15 ± 0.09 ^a^
EFP (%)	1.39 ± 0.23 ^a^	2.46 ± 0.62 ^c^	1.94 ± 0.47 ^b^	2.44 ± 0.22 ^c^	1.88 ± 0.53 ^b^
Heart (%)	0.51 ± 0.13 ^a^	0.72 ± 0.10 ^b^	0.73 ± 0.07 ^b^	0.79 ± 0.16 ^b^	0.77 ± 0.06 ^b^
Lung (%)	0.68 ± 0.14 ^a^	0.77 ± 0.10 ^a^	0.78 ± 0.09 ^a^	0.78 ± 0.14 ^a^	0.77 ± 0.19 ^a^
Brain (%)	1.74 ± 0.07 ^a^	1.88 ± 0.05 ^b^	1.88 ± 0.06 ^b^	1.84 ± 0.10 ^b^	1.88 ± 0.08 ^b^
BAT (%)	0.21 ± 0.03 ^a^	0.20 ± 0.04 ^a^	0.17 ± 0.03 ^a^	0.19 ± 0.04 ^a^	0.22 ± 0.07 ^a^

Data are mean ± SD for n = 8 mice in each group. Different superscript letters (a, b, c and d) indicate statistically significant differences among groups (*p* < 0.05, one-way ANOVA with Duncan’s post hoc test). Groups sharing the same letter are not significantly different. Final body weight values were measured after a 12-h fasting period immediately prior to euthanasia. Kidney weight refers to the total of both kidneys. YS, young sedentary; AS, aged sedentary; AE, aged with exercise; ASNMN, aged with NMN supplementation; AENMN, aged with combined NMN and exercise, BAT, brown adipose tissue; EFP, epididymal fat pad.

**Table 2 nutrients-17-03148-t002:** Effect of NMN supplementation and exercise training on the body composition in aging mice.

Characteristic	YS	AS	AE	ASNMN	AENMN
Body weight (g)	26.50 ± 1.33 ^a^	34.83 ± 2.37 ^b^	34.98 ± 1.29 ^b^	34.26 ± 2.25 ^b^	34.69 ± 2.03 ^b^
Fat mass (g)	3.19 ± 0.35 ^a^	4.50 ± 0.48 ^b^	4.24 ± 0.64 ^b^	4.30 ± 0.46 ^b^	4.45 ± 1.14 ^b^
Lean mass (g)	18.97 ± 1.09 ^a^	25.01 ± 1.65 ^b^	25.37 ± 0.77 ^b^	25.05 ± 1.84 ^b^	25.45 ± 1.48 ^b^
Free body fluid (g)	2.04 ± 0.15 ^a^	2.74 ± 0.20 ^b^	3.01 ± 0.59 ^b^	2.73 ± 0.19 ^b^	3.01 ± 0.24 ^b^

Data are mean ± SD for n = 8 mice in each group. Different superscript letters (a and b) indicate statistically significant differences among groups (*p* < 0.05, one-way ANOVA with Duncan’s post hoc test). Groups sharing the same letter are not significantly different. YS, young sedentary; AS, aged sedentary; AE, aged with exercise; ASNMN, aged with NMN supplementation; AENMN, aged with combined NMN and exercise.

**Table 3 nutrients-17-03148-t003:** Effect of NMN supplementation and exercise training on oral glucose tolerance test results.

OGTT	YS	AS	AE	ASNMN	AENMN
0 min (mg/dL)	130.88 ± 7.18 ^a^	133.13 ± 9.58 ^a^	130.88 ± 4.71 ^a^	128.75 ± 7.01 ^a^	131.25 ± 5.92 ^a^
15 min (mg/dL)	258.63 ± 42.70 ^a^	328.63 ± 42.70 ^b^	315.13 ± 71.21 ^ab^	309.38 ± 85.94 ^ab^	289.75 ± 48.13 ^ab^
30 min (mg/dL)	242.25 ± 36.56 ^bc^	267.75 ± 53.13 ^c^	229.13 ± 35.68 ^abc^	218.75 ± 52.90 ^ab^	197.25 ± 23.43 ^a^
60 min (mg/dL)	205.38 ± 26.75 ^b^	199.25 ± 26.95 ^b^	202.00 ± 19.41 ^b^	183.50 ± 45.96 ^ab^	159.38 ± 28.80 ^a^
120 min (mg/dL)	160.50 ± 13.23 ^bc^	167.63 ± 18.52 ^c^	159.63 ± 22.98 ^bc^	146.63 ± 18.16 ^ab^	137.25 ± 25.84 ^a^
AUC	25,538.44 ± 3093.39 ^ab^	25,544.06 ± 3791.65 ^b^	24,779.06 ± 3156.31 ^b^	23,123.44 ± 4195.80 ^ab^	21,075.00 ± 2685.25 ^a^

Data are mean ± SD for n = 8 mice in each group. Different superscript letters (a, b, and c) indicate statistically significant differences among groups (*p* < 0.05, one-way ANOVA with Duncan’s post hoc test). Groups sharing the same letter are not significantly different. YS, young sedentary; AS, aged sedentary; AE, aged with exercise; ASNMN, aged with NMN supplementation; AENMN, aged with combined NMN and exercise; AUC, area under the curve.

**Table 4 nutrients-17-03148-t004:** Effects of NMN supplementation and exercise training on biochemical parameters in aging mice.

Parameters	YS	AS	AE	ASNMN	AENMN
AST (U/L)	166.63 ± 21.0 ^a^	126.38 ± 60.35 ^a^	142.00 ± 54.82 ^a^	169.00 ± 67.07 ^a^	157.13 ± 97.44 ^a^
ALT (U/L)	35.50 ± 4.34 ^a^	39.63 ± 6 ^a^	39.38 ± 8 ^a^	41.13 ± 18 ^a^	46.75 ± 19 ^a^
CPK (U/L)	370.00 ± 74.73 ^a^	312.25 ± 174.18 ^a^	362.63 ± 200.42 ^a^	394.88 ± 205.50 ^a^	370.00 ± 362.49 ^a^
TP (g/dL)	5.96 ± 0.15 ^a^	5.73 ± 0.19 ^a^	5.81 ± 0.43 ^a^	5.78 ± 0.12 ^a^	5.80 ± 0.48 ^a^
ALB (g/dL)	3.70 ± 0.38 ^a^	3.63 ± 0.19 ^a^	3.70 ± 0.38 ^a^	3.65 ± 0.10 ^a^	3.64 ± 0.29 ^a^
BUN (mg/dL)	24.56 ± 1.75 ^a^	32.11 ± 4.82 ^b^	30.66 ± 3.66 ^b^	31.86 ± 4.30 ^b^	31.91 ± 7.80 ^b^
TC (mg/dL)	62.29 ± 2.93 ^a^	58.38 ± 7.42 ^a^	59.88 ± 8.34 ^a^	61.29 ± 4.90 ^a^	61.13 ± 10.86 ^a^
TG (mg/dL)	95.29 ± 13.39 ^b^	90.25 ± 14.68 ^b^	70.00 ± 9.13 ^a^	68.29 ± 6.82 ^a^	71.25 ± 11.91 ^a^
HDL-C (mg/dL)	70.38 ± 2.36 ^a^	68.28 ± 5.02 ^a^	73.18 ± 3.77 ^a^	72.89 ± 4.04 ^a^	75.23 ± 2.95 ^a^
LDL-C (mg/dL)	8.80 ± 0.95 ^a^	10.01 ± 1.61 ^a^	9.73 ± 1.63 ^a^	11.63 ± 0.63 ^a^	11.25 ± 3.22 ^a^
NAMPT(µg/mL)	7.49 ± 1.96 ^c^	3.49 ± 0.44 ^a^	5.63 ± 0.36 ^b^	5.68 ± 0.92 ^b^	6.07 ± 0.55 ^b^

Data are mean ± SD for n = 8 mice in each group. Different superscript letters (a, b, and c) indicate statistically significant differences among groups (*p* < 0.05, one-way ANOVA with Duncan’s post hoc test). Groups sharing the same letter are not significantly different. YS, young sedentary; AS, aged sedentary; AE, aged with exercise; ASNMN, aged with NMN supplementation; AENMN, aged with combined NMN and exercise; AST, aspartate transferase; ALT, alanine transferase; CPK, creatine phosphokinase; TP, total protein; ALB, albumin; BUN, blood urea nitrogen; TC, total cholesterol; TG, triglyceride; HDL-C, high-density lipoprotein cholesterol; LDL-C, low-density lipoprotein cholesterol; NAMPT, nicotinamide phosphoribosyltransferase.

## Data Availability

The original contributions presented in this study are included in the article. Further inquiries can be directed to the corresponding author.

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
