# Peer review of "Effects of Nicotinamide Mononucleotide Supplementation and Aerobic Exercise on Metabolic Health and Physical Performance in Aged Mice"

_nutrients, 2025, doi:10.3390/nu17193148_

Round 1
Reviewer 1 Report
Comments and Suggestions for Authors
This manuscript assesses the effects of NMN and exercise on metabolic health and physical performance in aged mice.
Overall, the manuscript is okay, and the study design and methods seem proper.
However, there are some discrepancies and missing information that impact the ability to adequately interpret the study results.
I have identified the following issues to be addressed in order to improve the manuscript:
Major issues:
- In the methods section, this study includes many different tests and evaluations. The way these are described is a bit confusing. The timeline of these tests is not clear. Which test was performed first, which followed, and how long did the authors wait between each test? Also, when were the different tests done, relative to the intervention beginning and ending? This information is very important in order to properly interpret the results of these tests. Please add this information in the methods section.
- Please include a flow chart or a diagram that clearly explains when did the intervention start, how long did it last and when were the different test done. It is crucial for proper interpretation of the results.
- In the methods section, body composition, when was this analysis done? Was it only done once or did the authors evaluate it pre and post intervention? Please clarify.
- In the measurement of metabolic rate, there were several issues here: a) please clarify when was this done? b) please clarify if it was only done once. c) the authors did not mention that this analysis involved spontaneous activity using running wheels. Please add this information.
- In the endurance exercise performance test, again, please clarify when was this done? how long into the intervention and how long before sacrificing the mice?
- In the weights test, please clarify when it was performed.
- In the weights test, please clarify how does this test compare to the grip strength test and why did the authors choose to use it here. How is it different that grip strength evaluation and what does it add to the evaluation of muscle strength?
- In the functional strength test, lines 215 – 216: “The setup employed was identical to 215 that used in the training protocol, allowing for consistent evaluation conditions” This is not clear. Please rephrase and clarify.
- In the Oral Glucose Tolerance Test, please clarify when this was done, how long before the sacrificing of the mice since the 12-hour fasting will affect many of the analyses performed.
- In the methods section the authors describe Pathophysiological Histology of Tissues Staining, lines 265 – 273. However, they never present the results of these analyses or even refer to them in the text. Please delete this section or include the results in the manuscript.
- In the results section, figure 1, If the intervention was for 6 weeks, why do the authors present bodyweights for 8 weeks? Were the animals not euthanized at the conclusion of 6 weeks? Please explain this discrepancy.
- Also in figure 1, can the changes in body weight in week 6 have to do with some of the analyses preformed, that involved fasting? It is not clear since the timing of the analysis and the timeline of the entire study is not specified in the methods.
- In Table 1, Please explain how these measurements were obtained (water intake, food intake, and caloric intake). Do they represent the entire intervention or only the 48 hours in the metabolic cages? Did these values change throughout the intervention? Are these values averages of different cages or of different mice? If so, how many mice, how many cages? None of this in mentioned in the methods!
- Also, please explain the units of measure: ml, or g, or kcal/mice/day, should it be ml/mouse/day?
- In the results, lines 375 – 384, and Figure 3, the authors describe wheel-running activity, but this was never mentioned in the methods! Please update the methods section to include a description of this measurement.
- In the results section, Effects of NMN Supplementation and Exercise Training on Oral Glucose Tolerance Test, since the authors do not specify WHEN was the OGTT done, how long after the last exercise session or NMN administration, it is impossible to determine if these are indeed effects of exercise training or acute effects of exercise. Please address this in the methods and the results sections.
- In the results section, Effects of NMN … on Endurance Performance…, and Figure 4. Here again, it is impossible to determine if these are acute effects since the timing of the testing is not specified.
- In the Western Blot description, lines 482- 489, and figure 7, the authors mention in the methods that the loading control is beta-actin but report in the results GAPDH. Which is it? GAPDH might be a bit problematic since it might be affected by aging and by metabolic intervention like exercise…. Please correct this and if GAPDH was used, address this in the discussion as a limitation of the study.
- In the discussion section, lines 506-508, the authors write: ‘This dual approach significantly enhanced aerobic capacity, muscular endurance, and, to a lesser extent, forelimb strength. Furthermore, improvements in glucose tolerance and increased expression of SirT1 in skeletal muscle were observed’.
This statement is mostly false. Aerobic capacity did not increase in AENMN more than in AE who received exercise alone, and evidence for changes in muscular endurance and strength is weak at best. Also, the authors data does not show significantly increased expression of Sirt1 in AENMN compared to AE. Please rephrase this part to a more conservative statement. - In the discussion, lines 522 – 526, the authors write: ‘Although some improvements in muscle strength and endurance were observed in specific intervention groups, the effects were limited and did not consistently reach statistical significance across all tests, suggesting that NMN supplementation and moderate aerobic exercise alone may have only modest benefits on neuromuscular performance in aging mice’. While this might be true, one must consider an alternative option in which this study lacks the statistical power to detect these changes. Many of the variables tested had very large variance, some even had SD that was larger than the mean.
- In the discussion, lines 532 – 572, the authors make several mechanistic speculations in an attempt to explain their findings. Many of these, like molecular markers of angiogenesis, capillary density, mitochondrial biogenesis, or mitochondrial content, and markers of glucose metabolism in the liver and muscle could, at least in part be evaluated to support these mechanistic statements. If possible, I would recommend performing at least some of these analyses with the paraffin embedded samples reported in the methods, or address this as a study limitation in the discussion.
- In the discussion, lines 573 – 583, the authors describe changes in response to NMN and exercise in their study but due to insufficient information about the timing of the experimental testing, the reader is left to wonder if these changes are indeed in response to exercise training and chronic administration of NMN or are they acute effects of the last exercise bout or the last NMN dose? It is crucial that the authors address this issue throughout the manuscript and especially in the method section: how long following the last bout of exercise and last dose of NMN testing was done and animals euthanized?
- In the discussion section, there is no mention of study limitations. I have made some suggestions throughout my report. Please include a short description of the study limitations.
Minor issues:
- In the introduction, line 58, please change ‘exercise’ to ‘regular exercise’ or ‘exercise training’
- In the introduction, lines 78-80, if NMN is converted to NR, which is also available as a supplement, please explain why the authors decided to use NMN and not NR. What are the advantages of using NMN instead of NR?
- In the methods section, line 127, 6 weeks of intervention, although adequate, might be too short to induce detectable muscle adaptations and might have impacted the degree of the effects of exercise detected in this study. This should be mentioned in the discussion section, as a possible explanation or at the very least as a study limitation.
- In the methods sections, lines 127-128, please clarify if mice were individually housed or groups housed and how many mice per cage and if young mice were housed separately from aged mice.
- In the methods section, line 129, why did the authors use distilled water for the mice?
- In the methods section, line 130, please include the dietary composition of the diet.
- In the exercise intervention protocol, line 140, please explain why 6 weeks was chosen for the intervention duration?
- In the exercise intervention protocol, line 146, why 20 min? seems too short. Also, if the mice started at 10 min and gradually increased the running time, at which week the 20 min duration was achieved?
- In the exercise intervention protocol, was electrical shock used as a negative motivator? Please clarify and state it clearly in this section.
- In the Measurement of Biochemical Parameters, line 246, the information of the ELISA kit is incorrect and specifies Visfatin/PBEF as the protein detected. Please correct it to the proper (NAMPT) kit.
- In the Western Blotting section, line 249, please clarify if the skeletal muscles were harvested from the hind limb or the forelimb.
- In the results, lines 313-314, ‘we divided tissue weight by a relative percentage of body weight and found similar results to absolute tissue weight.’ Should probably be ‘we divided tissue weight by body weight to calculate the relative percentage of body weight and found similar results to absolute tissue weight.’
- In the result section, lines 315 – 323 and Table 2, could the lack of difference in body composition be possibly explained by a relative short intervention duration (6 weeks only)?
- In the result section (and also in the methods), the term Respiratory exchange ratio (RER)is more appropriate here instead of respiratory quotient (RQ).
- In figure 2, please correct the X axis and change ‘Night’ to ‘Dark’ and change ‘Fully Day’ to ’24 Hours’.
- In table 4, please explain why the LDL value in the AENMN group has such large variance (SD larger than the mean).
- In figure 7, please correct the title, move the letters ‘Fig’ to line 491.
- In the discussion, line 538, remove the ‘i’ from ‘endurancei’.
- In the discussion, line 576, change ‘fat metabolism’ to ‘fat oxidation’.
In summary, this manuscript addresses an important topic and can be improved by applying the above recommendations. Particularly these related to reporting the study design/methods and interpretation of the results.
Author Response
" Effects of Nicotinamide Mononucleotide Supplementation and Aerobic Exercise on Metabolic Health and Physical Performance in Aged Mice (nutrients-3902121)"
Response to Reviewer’s Comments
For Reviewer #1 :
This manuscript assesses the effects of NMN and exercise on metabolic health and physical performance in aged mice. Overall, the manuscript is okay, and the study design and methods seem proper. However, there are some discrepancies and missing information that impact the ability to adequately interpret the study results.
Response: We sincerely thank Reviewer #1 for the thoughtful and constructive comments on our manuscript. Your insights have been invaluable in helping us enhance the scientific rigor and clarity of our work. In response, we have undertaken a comprehensive revision of the manuscript, with specific improvements in methodological descriptions, data interpretation, statistical reporting, and overall language quality. We have also carefully refined the discussion to provide a balanced and appropriately cautious interpretation of the findings, consistent with the presented results. All revisions have been highlighted in red in the revised manuscript for ease of review. We greatly appreciate your valuable input and sincerely hope that the revised version satisfactorily addresses all your concerns.
Major issues:
- In the methods section, this study includes many different tests and evaluations. The way these are described is a bit confusing. The timeline of these tests is not clear. Which test was performed first, which followed, and how long did the authors wait between each test? Also, when were the different tests done, relative to the intervention beginning and ending? This information is very important in order to properly interpret the results of these tests. Please add this information in the methods section.
Response: We thank the reviewer for highlighting this important issue. In the revised Methods (Section 2.1), we have added a detailed description of the experimental timeline. Specifically, after completion of the 6-week intervention, functional performance tests , including endurance treadmill running, grip strength, weight-loading test, Kondziela’s inverted screen, functional strength, and rotarod. These were followed by OGTT and indirect calorimetry. After all in vivo measurements, animals were euthanized for blood biochemical analyses, tissue collection, histological examination, and molecular assays. For clarity, we have also included a schematic experimental timeline (Figure 1).
- Please include a flow chart or a diagram that clearly explains when did the intervention start, how long did it last and when were the different test done. It is crucial for proper interpretation of the results.
Response: We thank the reviewer for this helpful suggestion. In the revised manuscript, we have added a schematic diagram (Figure 1) that illustrates the experimental timeline, including the start and 6-week duration of the intervention, as well as the subsequent sequence of functional performance tests, oral glucose tolerance test (OGTT), indirect calorimetry, and sacrifice. We believe this figure provides a clear overview of the study design and greatly facilitates interpretation of the results.
- In the methods section, body composition, when was this analysis done? Was it only done once or did the authors evaluate it pre and post intervention? Please clarify.
Response: We sincerely thank the reviewer for raising this important point. In our study, body composition was assessed only once, on Day 43, immediately after completion of the 6-week intervention (Day 42). Pre- and post-intervention comparisons were not conducted. This clarification has been added to the Materials and Methods section (Section 2.4) and is also illustrated in Figure 1 for clarity.
- In the measurement of metabolic rate, there were several issues here: a) please clarify when was this done? b) please clarify if it was only done once. c) the authors did not mention that this analysis involved spontaneous activity using running wheels. Please add this information.
Response: We appreciate the reviewer’s careful comments on this section. To clarify, whole-body metabolic rate and associated physiological parameters were measured once on Days 53–55, after completion of the 6-week intervention (Day 42). The assessment was performed over a 48-h recording period following a 24-h acclimation. In addition, as suggested, we have now specified that spontaneous locomotor activity was also evaluated, using wheel-running activity integrated into the calorimetry system. These clarifications have been added to the revised Materials and Methods (Section 2.12) and are also depicted in Figure 1.
- In the endurance exercise performance test, again, please clarify when was this done? how long into the intervention and how long before sacrificing the mice?
Response: We thank the reviewer for this helpful comment. To clarify, the treadmill endurance test was performed once on Day 44, two days after completion of the 6-week intervention (Day 42), and approximately 12 days before sacrifice on Day 56. This clarification has been added to the revised Materials and Methods (Section 2.5) and is also illustrated in Figure 1.
- In the weights test, please clarify when it was performed.
Response: We appreciate the reviewer’s comment. The weights test was performed once on Day 46, four days after completion of the 6-week intervention (Day 42), and approximately 10 days before sacrifice on Day 56. This clarification has been added to the revised Materials and Methods (Section 2.7) and is also illustrated in Figure 1.
- In the weights test, please clarify how does this test compare to the grip strength test and why did the authors choose to use it here. How is it different that grip strength evaluation and what does it add to the evaluation of muscle strength?
Response: We thank the reviewer for this insightful comment. The grip strength test and the weights test were intentionally included as complementary methods, as they capture different aspects of neuromuscular performance in aged mice. The grip strength test measures maximal voluntary forelimb force in a single pulling action, reflecting instantaneous maximal strength. In contrast, the weights test requires mice to voluntarily grasp and lift progressively heavier weighted balls, thereby assessing functional strength and muscular endurance under increasing resistance.
Together, these tests provide a more comprehensive evaluation: grip strength reflects peak force generation, whereas the weights test offers insight into sustained performance under load. Importantly, studies in aged mice have demonstrated that aging is associated not only with reduced maximal grip force but also with impaired endurance and resistance to fatigue, underscoring the value of combining both approaches (Takeshita et al., 2017; Deacon, 2013). In addition, standard operating procedures and methodological reviews recognize the weights test as a reliable tool to assess muscle performance in rodents (Deacon, 2013; Owendoff et al., 2023). For these reasons, we employed both tests to strengthen the validity of our neuromuscular assessment in aging mice.
References
Deacon R. M. (2013). Measuring the strength of mice. Journal of visualized experiments : JoVE, (76), 2610. https://doi.org/10.3791/2610
Owendoff, G., Ray, A., Bobbili, P., Clark, L., Baumann, C. W., Clark, B. C., & Arnold, W. D. (2023). Optimization and construct validity of approaches to preclinical grip strength testing. Journal of cachexia, sarcopenia and muscle, 14(5), 2439–2445. https://doi.org/10.1002/jcsm.13300
Takeshita, H., Yamamoto, K., Nozato, S., Inagaki, T., Tsuchimochi, H., Shirai, M., Yamamoto, R., Imaizumi, Y., Hongyo, K., Yokoyama, S., Takeda, M., Oguro, R., Takami, Y., Itoh, N., Takeya, Y., Sugimoto, K., Fukada, S. I., & Rakugi, H. (2017). Modified forelimb grip strength test detects aging-associated physiological decline in skeletal muscle function in male mice. Scientific reports, 7, 42323. https://doi.org/10.1038/srep42323
- In the functional strength test, lines 215 – 216: “The setup employed was identical to 215 that used in the training protocol, allowing for consistent evaluation conditions” This is not clear. Please rephrase and clarify.
Response: We thank the reviewer for pointing out this ambiguity. The original sentence has been rephrased for clarity. Specifically, we now state that the same apparatus and mesh used during the training sessions were also employed in the functional strength test to ensure consistent evaluation conditions and minimize variability. This revision has been incorporated into the Materials and Methods (Section 2.9).
- In the Oral Glucose Tolerance Test, please clarify when this was done, how long before the sacrificing of the mice since the 12-hour fasting will affect many of the analyses performed.
Response: We thank the reviewer for raising this important point. To clarify, the oral glucose tolerance test (OGTT) was performed once on Day 51, after a 12-h overnight fast, and five days before sacrifice on Day 56. This interval allowed sufficient recovery from the fasting period and minimized any potential carry-over effects on subsequent biochemical and tissue analyses. This clarification has been added to the revised Materials and Methods (Section 2.11) and is also illustrated in Figure 1.
- In the methods section the authors describe Pathophysiological Histology of Tissues Staining, lines 265 – 273. However, they never present the results of these analyses or even refer to them in the text. Please delete this section or include the results in the manuscript.
Response: We thank the reviewer for pointing this out. This section was mistakenly included in the original submission, and we apologize for the oversight. It has now been removed from the revised manuscript.
- In the results section, figure 1, If the intervention was for 6 weeks, why do the authors present bodyweights for 8 weeks? Were the animals not euthanized at the conclusion of 6 weeks? Please explain this discrepancy.
Response: We thank the reviewer for noticing this discrepancy. To clarify, the intervention itself lasted for 6 weeks (Day 0-42). However, body weight was continuously monitored until the end of the experiment (Day 56), which included the additional period when the functional performance tests, OGTT, and indirect calorimetry were performed prior to sacrifice. Accordingly, Figure 1 presents 8 weeks of body weight data to reflect the entire experimental timeline rather than only the intervention phase. The figure legend and corresponding text in the Results section have been revised to clearly indicate this distinction.
- Also in figure 1, can the changes in body weight in week 6 have to do with some of the analyses preformed, that involved fasting? It is not clear since the timing of the analysis and the timeline of the entire study is not specified in the methods.
Response: We thank the reviewer for this valuable comment. To clarify, body weight was recorded routinely throughout the 6-week intervention and the subsequent testing period (up to Day 56). The observed changes in body weight during week 6 were not attributable to fasting procedures, as fasting was implemented only later for the OGTT (Day 51, i.e., week 7) and immediately prior to sacrifice (Day 56). To avoid confusion, we have revised the Materials and Methods (Section 2.1) and the Figure 1 legend to more clearly specify the timing of the fasting procedures and the overall experimental timeline.
- In Table 1, Please explain how these measurements were obtained (water intake, food intake, and caloric intake). Do they represent the entire intervention or only the 48 hours in the metabolic cages? Did these values change throughout the intervention? Are these values averages of different cages or of different mice? If so, how many mice, how many cages? None of this in mentioned in the methods!
Response: We thank the reviewer for this comment and the opportunity to clarify. The data for water, food, and caloric intake in Table 1 were obtained over the entire 6-week intervention period. Water bottles and food pellets were weighed weekly for each cage, and consumption was calculated as the difference between the initial and final weights, corrected for spillage. These amounts were then normalized to the number of mice per cage and averaged across all cages to yield the mean daily intake per mouse. Caloric intake was derived from the manufacturer’s nutritional composition of the chow diet (PMI 5001). Thus, Table 1 reflects the average daily intake per mouse across the full 6-week intervention. This clarification has been added to the Materials and Methods (Section 2.1).
- Also, please explain the units of measure: ml, or g, or kcal/mice/day, should it be ml/mouse/day?
Response: We thank the reviewer for pointing this out. To improve clarity, the units in Table 1 have been corrected from ml/mice/day, g/mice/day, and kcal/mice/day to ml/mouse/day, g/mouse/day, and kcal/mouse/day, thereby accurately indicating intake on a per-mouse basis.
- In the results, lines 375 – 384, and Figure 3, the authors describe wheel-running activity, but this was never mentioned in the methods! Please update the methods section to include a description of this measurement.
Response: We thank the reviewer for highlighting this omission. We have now updated the Materials and Methods (Section 2.12) to include a detailed description of running wheel activity, specifying that spontaneous locomotor activity was recorded continuously during the indirect calorimetry assessment using wheel-running sensors integrated into the system.
- In the results section, Effects of NMN Supplementation and Exercise Training on Oral Glucose Tolerance Test, since the authors do not specify WHEN was the OGTT done, how long after the last exercise session or NMN administration, it is impossible to determine if these are indeed effects of exercise training or acute effects of exercise. Please address this in the methods and the results sections.
Response: We thank the reviewer for this important comment. The timing of the OGTT has now been clarified in the Materials and Methods (Section 2.11), where we specify that the test was performed on Day 51, after completion of the 6-week intervention. In the Results section, we now simply describe the findings as occurring “after the 6-week intervention” to maintain clarity and avoid redundancy.
- In the results section, Effects of NMN … on Endurance Performance…, and Figure 4. Here again, it is impossible to determine if these are acute effects since the timing of the testing is not specified.
Response: We thank the reviewer for this helpful comment. The treadmill endurance test was performed on Day 44, two days after completion of the 6-week intervention, thereby ensuring that the results reflect the effects of the intervention rather than acute exercise. This clarification has been added to the Materials and Methods (Section 2.5) and noted in the corresponding Results section.
- In the Western Blot description, lines 482- 489, and figure 7, the authors mention in the methods that the loading control is beta-actin but report in the results GAPDH. Which is it? GAPDH might be a bit problematic since it might be affected by aging and by metabolic intervention like exercise…. Please correct this and if GAPDH was used, address this in the discussion as a limitation of the study.
Response: We thank the reviewer for carefully noting this issue. In the Materials and Methods section, β-actin was incorrectly stated as the loading control. In fact, GAPDH was used as the internal loading control in our experiments. This has now been corrected in the Materials and Methods (Section 2.15) to ensure consistency with the Results (Section 3.8 and Figure 8). We would like to emphasize that GAPDH was applied solely as a normalization control for protein loading, not as a target protein of interest. As such, potential variability in GAPDH expression does not affect the interpretation of our target protein results. We apologize for the earlier inaccuracy and respectfully ask for the reviewer’s understanding.
- In the discussion section, lines 506-508, the authors write: ‘This dual approach significantly enhanced aerobic capacity, muscular endurance, and, to a lesser extent, forelimb strength. Furthermore, improvements in glucose tolerance and increased expression of SirT1 in skeletal muscle were observed’.
This statement is mostly false. Aerobic capacity did not increase in AENMN more than in AE who received exercise alone, and evidence for changes in muscular endurance and strength is weak at best. Also, the authors data does not show significantly increased expression of Sirt1 in AENMN compared to AE. Please rephrase this part to a more conservative statement.
Response: We thank the reviewer for this important observation. To avoid overstating the findings, we have rephrased the relevant sentences in the Conclusions section to provide a more conservative interpretation. Specifically, we now emphasize that the combined NMN and exercise intervention improved aerobic endurance, glucose tolerance, and systemic metabolism, whereas the effects on muscular endurance, strength, and SirT1 expression were modest and not consistently greater than those achieved by exercise alone.
- In the discussion, lines 522 – 526, the authors write: ‘Although some improvements in muscle strength and endurance were observed in specific intervention groups, the effects were limited and did not consistently reach statistical significance across all tests, suggesting that NMN supplementation and moderate aerobic exercise alone may have only modest benefits on neuromuscular performance in aging mice’. While this might be true, one must consider an alternative option in which this study lacks the statistical power to detect these changes. Many of the variables tested had very large variance, some even had SD that was larger than the mean.
Response: We appreciate the reviewer’s thoughtful comment. We agree that the absence of consistent statistical significance may also reflect the limited sample size and relatively high variability observed in some outcome measures, which could have reduced statistical power to detect smaller effects. To address this, we have revised the Discussion section to acknowledge this possibility and note that larger cohorts will be necessary in future studies to more reliably evaluate the effects of NMN and exercise on neuromuscular performance.
- In the discussion, lines 532 – 572, the authors make several mechanistic speculations in an attempt to explain their findings. Many of these, like molecular markers of angiogenesis, capillary density, mitochondrial biogenesis, or mitochondrial content, and markers of glucose metabolism in the liver and muscle could, at least in part be evaluated to support these mechanistic statements. If possible, I would recommend performing at least some of these analyses with the paraffin embedded samples reported in the methods, or address this as a study limitation in the discussion.
Response: We thank the reviewer for this valuable suggestion. In the revised Discussion (4th-5th paragraphs, previously 4th–6th), we have rephrased the text in a more conservative tone and explicitly acknowledged that we did not directly assess angiogenesis, capillary density, mitochondrial biogenesis, or glucose metabolism markers. While additional analyses of paraffin-embedded samples could indeed provide further mechanistic insights, these experiments were beyond the scope of the present study. Accordingly, the mechanistic statements are now clearly indicated as speculative and are discussed as a limitation. We will also take this important recommendation into account when designing future studies.
- In the discussion, lines 573 – 583, the authors describe changes in response to NMN and exercise in their study but due to insufficient information about the timing of the experimental testing, the reader is left to wonder if these changes are indeed in response to exercise training and chronic administration of NMN or are they acute effects of the last exercise bout or the last NMN dose? It is crucial that the authors address this issue throughout the manuscript and especially in the method section: how long following the last bout of exercise and last dose of NMN testing was done and animals euthanized?
Response: We thank the reviewer for raising this important point. To address this concern, we have supplemented the Methods with detailed information regarding the timing of each experimental test relative to the last bout of exercise and the final NMN administration. Specifically, all testing was conducted after completion of the 6-week intervention: the treadmill endurance test was performed on Day 44 (two days after the final intervention), the weights test on Day 46 (four days after), the OGTT on Day 51 (nine days after, following a 12-h fast), indirect calorimetry on Days 53–55 (11-13 days after), and euthanasia with tissue collection on Day 56 (14 days after). In addition, we have added a schematic experimental timeline (Figure 1) to provide a clearer overview of the study design and testing schedule. These revisions clarify that the measurements were not conducted under acute effects of the last exercise session or NMN dose.
- In the discussion section, there is no mention of study limitations. I have made some suggestions throughout my report. Please include a short description of the study limitations.
Response: We thank the reviewer for this important suggestion. In line with the comments provided, we have now added a brief description of study limitations in the Discussion section. Specifically, we note that the relatively small sample size and 6-week intervention duration may limit the detection of subtle or long-term adaptations, and that mechanistic analyses such as angiogenesis, mitochondrial biogenesis, or tissue-specific metabolic markers were not performed. These points are now acknowledged as limitations, and we have emphasized that future studies with larger cohorts, extended intervention periods, and more comprehensive mechanistic evaluations will be necessary to further validate and expand upon our findings.
Minor issues:
- In the introduction, line 58, please change ‘exercise’ to ‘regular exercise’ or ‘exercise training’
Response: Thank you for the helpful suggestion. We have revised “exercise” to “exercise training” in line 58 of the Introduction for greater precision.
- In the introduction, lines 78-80, if NMN is converted to NR, which is also available as a supplement, please explain why the authors decided to use NMN and not NR. What are the advantages of using NMN instead of NR?
Response: We appreciate the reviewer’s question. We selected NMN rather than NR because oral NMN, although partly dephosphorylated to NR with subsequent re-phosphorylation in enterocytes, is also reported to use direct intestinal uptake pathways, indicating overlapping yet distinct handling relative to NR (Grozio et al., 2019; Schmidt & Brenner, 2019). In human studies, oral NMN has produced consistent increases in circulating NAD⁺, whereas NR has shown greater variability in target tissues despite frequent elevations in blood NAD⁺ (Okabe et al., 2022; Elhassan et al., 2019). These considerations provided a clear mechanistic and pharmacodynamic rationale for our study.
References
Elhassan, Y. S., Kluckova, K., Fletcher, R. S., Schmidt, M. S., Garten, A., Doig, C. L., Cartwright, D. M., Oakey, L., Burley, C. V., Jenkinson, N., Wilson, M., Lucas, S. J. E., Akerman, I., Seabright, A., Lai, Y. C., Tennant, D. A., Nightingale, P., Wallis, G. A., Manolopoulos, K. N., Brenner, C., Philp, A., Lavery, G. G. (2019). Nicotinamide Riboside Augments the Aged Human Skeletal Muscle NAD+ Metabolome and Induces Transcriptomic and Anti-inflammatory Signatures. Cell reports, 28(7), 1717–1728.e6. https://doi.org/10.1016/j.celrep.2019.07.043
Grozio, A., Mills, K. F., Yoshino, J., Bruzzone, S., Sociali, G., Tokizane, K., Lei, H. C., Cunningham, R., Sasaki, Y., Migaud, M. E., & Imai, S. I. (2019). Slc12a8 is a nicotinamide mononucleotide transporter. Nature metabolism, 1(1), 47–57. https://doi.org/10.1038/s42255-018-0009-4
Okabe, K., Yaku, K., Uchida, Y., Fukamizu, Y., Sato, T., Sakurai, T., Tobe, K., & Nakagawa, T. (2022). Oral Administration of Nicotinamide Mononucleotide Is Safe and Efficiently Increases Blood Nicotinamide Adenine Dinucleotide Levels in Healthy Subjects. Frontiers in nutrition, 9, 868640. https://doi.org/10.3389/fnut.2022.868640
Schmidt, M. S., & Brenner, C. (2019). Absence of evidence that Slc12a8 encodes a nicotinamide mononucleotide transporter. Nature metabolism, 1(7), 660–661. https://doi.org/10.1038/s42255-019-0085-0
- In the methods section, line 127, 6 weeks of intervention, although adequate, might be too short to induce detectable muscle adaptations and might have impacted the degree of the effects of exercise detected in this study. This should be mentioned in the discussion section, as a possible explanation or at the very least as a study limitation.
Response: We appreciate this valuable comment. In the revised Discussion (third paragraph), we now note that the limited effects observed may be attributable to the relatively short 6-week intervention period, which may not have been sufficient to induce detectable skeletal muscle adaptations. This point is now acknowledged as a possible explanation and a limitation of the study.
- In the methods sections, lines 127-128, please clarify if mice were individually housed or groups housed and how many mice per cage and if young mice were housed separately from aged mice.
Response: We appreciate the reviewer’s comment. The mice were group-housed, with four animals per cage. To avoid potential confounding effects, animals from different experimental groups (e.g., young vs. aged or sedentary vs. intervention) were not housed together.
- In the methods section, line 129, why did the authors use distilled water for the mice?
Response: We appreciate the reviewer’s question. Distilled water was provided as drinking water, in line with the standard practice of our animal facility. According to the Guide for the Care and Use of Laboratory Animals (National Research Council, 2011), potable and uncontaminated water should be available to animals at all times. The use of distilled water is widely adopted in rodent studies to ensure consistent water quality and to avoid variability introduced by minerals or additives present in tap water (Su et al., 2023; Kim et al., 2021).
References
Su, L. Y., Huang, W. C., Kan, N. W., Tung, T. H., Huynh, L. B. P., & Huang, S. Y. (2023). Effects of Resveratrol on Muscle Inflammation, Energy Utilisation, and Exercise Performance in an Eccentric Contraction Exercise Mouse Model. Nutrients, 15(1), 249. https://doi.org/10.3390/nu15010249
Kim, S., Kim, K., Park, J., & Jun, W. (2021). Curcuma longa L. Water Extract Improves Dexamethasone-Induced Sarcopenia by Modulating the Muscle-Related Gene and Oxidative Stress in Mice. Antioxidants (Basel, Switzerland), 10(7), 1000. https://doi.org/10.3390/antiox10071000
- In the methods section, line 130, please include the dietary composition of the diet.
Response: We appreciate the reviewer’s comment. The basic dietary composition of the standard chow diet has now been added in Methods section 2.1.
- In the exercise intervention protocol, line 140, please explain why 6 weeks was chosen for the intervention duration?
Response: We appreciate the reviewer’s comment. The 6-week duration was selected based on numerous previous studies using aged mice, where interventions of this length are commonly adopted and sufficient to elicit measurable physiological and metabolic adaptations while minimizing the risk of age-related morbidity during prolonged protocols (Lee et al., 2019; Kinoshita et al., 2021). Moreover, extending the intervention further may increase the likelihood of age-related morbidity or mortality, which could compromise both study completion and data integrity.
References
Lee, M., Oikawa, S., Ushida, T., Suzuki, K., & Akimoto, T. (2019). Effects of Exercise Training on Growth and Differentiation Factor 11 Expression in Aged Mice. Frontiers in physiology, 10, 970. https://doi.org/10.3389/fphys.2019.00970
Kinoshita, K., Hamanaka, G., Ohtomo, R., Takase, H., Chung, K. K., Lok, J., Lo, E. H., Katsuki, H., & Arai, K. (2021). Mature Adult Mice With Exercise-Preconditioning Show Better Recovery After Intracerebral Hemorrhage. Stroke, 52(5), 1861–1865. https://doi.org/10.1161/STROKEAHA.120.032201
- In the exercise intervention protocol, line 146, why 20 min? seems too short. Also, if the mice started at 10 min and gradually increased the running time, at which week the 20 min duration was achieved?
Response: We appreciate the reviewer’s comment. The exercise protocol was designed as a progressive regimen: mice began with 10 minutes during the acclimation phase, and the running duration was gradually increased each week to reach 20 minutes by week 6. The choice of 20 minutes was guided by previous studies in aged rodents, where similar treadmill protocols have been applied and shown to be both feasible and effective (Lovatel et al., 2013; Friedman & Kohn, 2022). To address this point, we have revised the Materials and Methods (Section 2.3).
References
Lovatel, G. A., Elsner, V. R., Bertoldi, K., Vanzella, C., Moysés, F.dosS., Vizuete, A., Spindler, C., Cechinel, L. R., Netto, C. A., Muotri, A. R., & Siqueira, I. R. (2013). Treadmill exercise induces age-related changes in aversive memory, neuroinflammatory and epigenetic processes in the rat hippocampus. Neurobiology of learning and memory, 101, 94–102. https://doi.org/10.1016/j.nlm.2013.01.007
Friedman, M. A., & Kohn, D. H. (2022). Calcium and phosphorus supplemented diet increases bone volume after thirty days of high speed treadmill exercise in adult mice. Scientific reports, 12(1), 14616. https://doi.org/10.1038/s41598-022-19016-8
- In the exercise intervention protocol, was electrical shock used as a negative motivator? Please clarify and state it clearly in this section.
Response:We thank the reviewer for this important question. A mild electrical stimulus was indeed used during treadmill training. Following established rodent treadmill protocols, the shock grid was set to deliver 0.2 mA, a parameter commonly employed in both adult and aged mouse studies. This level of stimulation has been shown to be aversive but not injurious, and is widely accepted as an effective motivator in treadmill exercise experiments (E et al., 2014; Leuchtmann et al., 2023). This clarification has now been explicitly added to the Materials and Methods (Section 2.3).
References
E, L., Burns, J. M., & Swerdlow, R. H. (2014). Effect of high-intensity exercise on aged mouse brain mitochondria, neurogenesis, and inflammation. Neurobiology of aging, 35(11), 2574–2583. https://doi.org/10.1016/j.neurobiolaging.
Leuchtmann, A. B., Afifi, Y., Ritz, D., & Handschin, C. (2023). Effects of high-resistance wheel running on hallmarks of endurance and resistance training adaptations in mice. Physiological reports, 11(11), e15701. https://doi.org/10.14814/phy2.15701
- In the Measurement of Biochemical Parameters, line 246, the information of the ELISA kit is incorrect and specifies Visfatin/PBEF as the protein detected. Please correct it to the proper (NAMPT) kit.
Response: We thank the reviewer for pointing this out. The ELISA kit information has been corrected to the NAMPT (visfatin/PBEF) Mouse/Rat Dual ELISA Kit (AdipoGen Life Sciences, catalog AG-45A-0007YEK-KI01), which specifically detects mouse/rat NAMPT. We have also standardized the nomenclature to NAMPT throughout the manuscript (Methods Section 2.14 ) to avoid confusion.
- In the Western Blotting section, line 249, please clarify if the skeletal muscles were harvested from the hind limb or the forelimb.
Response: We thank the reviewer for this helpful comment. We confirm that the skeletal muscle samples used for Western blotting were collected from the hind limb (gastrocnemius and soleus muscles). This clarification has been added to the Materials and Methods (Section 2.15).
- In the results, lines 313-314, ‘we divided tissue weight by a relative percentage of body weight and found similar results to absolute tissue weight.’ Should probably be ‘we divided tissue weight by body weight to calculate the relative percentage of body weight and found similar results to absolute tissue weight.’
Response: We agree with the reviewer’s suggestion. The sentence has been corrected to: “Because tissue weight may be influenced by overall body weight, we divided tissue weight by body weight to calculate the relative percentage of body weight, and found similar results to absolute tissue weight.”
- In the result section, lines 315 – 323 and Table 2, could the lack of difference in body composition be possibly explained by a relative short intervention duration (6 weeks only)?
Response: We agree with the reviewer’s comment. The absence of significant differences in body composition among the aged groups may indeed be attributed to the relatively short intervention period (6 weeks), which may not have been sufficient to elicit measurable changes in fat and lean mass. This possible explanation has now been acknowledged in the Discussion (third paragraph).
- In the result section (and also in the methods), the term Respiratory exchange ratio (RER)is more appropriate here instead of respiratory quotient (RQ).
Response: We thank the reviewer for this helpful comment. We agree that the term respiratory exchange ratio (RER) is more appropriate in the context of indirect calorimetry, as it reflects the VO₂ and VCO₂ measurements obtained under experimental conditions. Accordingly, all mentions of respiratory quotient (RQ) in the Methods (Section 2.12), Results, and Figure 3 have been corrected to respiratory exchange ratio (RER) in the revised manuscript.
- In figure 2, please correct the X axis and change ‘Night’ to ‘Dark’ and change ‘Fully Day’ to ’24 Hours’.
Response: We thank the reviewer for pointing out this issue. As suggested, the labeling on the X axis in Figure 2 has been corrected: Night has been changed to Dark, and Fully Day has been replaced with 24 Hours. These revisions have been implemented in the updated figure to improve accuracy and clarity.
- In table 4, please explain why the LDL value in the AENMN group has such large variance (SD larger than the mean).
Response: We thank the reviewer for noticing this discrepancy. After rechecking the raw data, we identified a transcription error in Table 4, where the HDL-C and LDL-C values for the AENMN group were mistakenly interchanged. The corrected values have now been updated in Table 4, and the statistical outcomes remain unchanged. We sincerely apologize for this oversight.
- In figure 7, please correct the title, move the letters ‘Fig’ to line 491.
Response: We thank the reviewer for pointing this out. The figure title has been corrected, and the label “Fig” has been properly repositioned in the revised manuscript.
- In the discussion, line 538, remove the ‘i’ from ‘endurancei’.
Response: We thank the reviewer for pointing out this typographical error. It has been corrected in the revised manuscript.
- In the discussion, line 576, change ‘fat metabolism’ to ‘fat oxidation’.
Response: We thank the reviewer for this helpful suggestion. The term “fat metabolism” has been revised to “fat oxidation” in the Discussion.

Reviewer 2 Report
Comments and Suggestions for Authors
Overall Assessment
This study investigates an important question about combining NMN supplementation with aerobic exercise to combat age-related decline. The research addresses a genuine gap in knowledge regarding potential synergistic effects between nutritional and exercise interventions in aging populations.
Strengths:
Well-controlled experimental design with appropriate control groups
Comprehensive assessment of multiple outcome measures (metabolic, physical performance, biochemical)
Appropriate statistical methodology using one-way ANOVA with Duncan's post-hoc test
Proper animal ethics approval and care protocols
Relevant aging model using 85-week-old mice
Weaknesses:
Small sample size (n=8 per group) may limit statistical power
Short intervention period (6 weeks) for aging research
Limited mechanistic insights beyond SirT1 and NAMPT measurements
Missing key reliability testing for physical performance measures
Inconsistent decimal formatting throughout the manuscript
Critical Areas for Improvement
1. Statistical Power and Sample Size (Methods section, page 3, lines 120-132)
The sample size appears insufficient for detecting meaningful differences. The authors should calculate and report the statistical power for their primary outcomes. For aging research with multiple comparisons, larger sample sizes (n=10-12 per group) would strengthen the conclusions.
2. Reliability and Validity of Measurements (Methods sections, pages 4-6)
The study lacks test-retest reliability data for physical performance measures, which is crucial for aging research where performance can be variable. This aligns with systematic review guidelines requiring robust measurement protocols. The authors should include reliability coefficients for their physical performance tests.
3. Decimal Formatting Issues
Throughout the manuscript, inconsistent decimal formatting appears (e.g., using commas instead of periods in some instances). All decimal numbers should use periods as separators and maintain at least two decimal places for precision.
4. Western Blot Sample Size (Figure 7, page 16, line 492)
The Western blot analysis used only n=2 per group, which is inadequate for meaningful statistical comparisons. This should be increased to at least n=6-8 per group to match other analyses.
Suggested Reference Enhancements
Specific Technical Corrections
Figure Quality and Error Bars (Figures 1-7)
All figures appropriately include error bars (standard deviation), which is commendable. However, Figure 7's Western blot quantification should include larger sample sizes as mentioned above.
Statistical Analysis Verification
The statistical approach is generally sound, using appropriate one-way ANOVA with post-hoc testing. However, the authors should consider:
Bonferroni correction for multiple comparisons
Effect size reporting alongside p-values
Confidence intervals for key findings
Methodological Considerations
The study design follows appropriate principles but would benefit from:
Longer intervention duration (12+ weeks) for aging research
Additional mechanistic markers beyond SirT1/NAMPT
More comprehensive body composition analysis
Larger sample sizes for adequate statistical power
Conclusion
This study addresses an important research question and provides valuable preliminary data on NMN-exercise interactions in aging. However, several methodological limitations constrain the strength of conclusions. The suggested improvements, particularly regarding sample sizes, measurement reliability, and enhanced statistical rigor, would significantly strengthen the manuscript's scientific contribution.
The manuscript demonstrates good scientific English overall, though some sections could benefit from refinement. For instance, page 16, lines 506-508 contains awkward phrasing that should be restructured for clarity.
Methodological Considerations
Author Response
" Effects of Nicotinamide Mononucleotide Supplementation and Aerobic Exercise on Metabolic Health and Physical Performance in Aged Mice (nutrients-3902121)"
Response to Reviewer’s Comments
For Reviewer #2:
Overall Assessment
This study investigates an important question about combining NMN supplementation with aerobic exercise to combat age-related decline. The research addresses a genuine gap in knowledge regarding potential synergistic effects between nutritional and exercise interventions in aging populations.
Strengths:
Well-controlled experimental design with appropriate control groups
Comprehensive assessment of multiple outcome measures (metabolic, physical performance, biochemical)
Appropriate statistical methodology using one-way ANOVA with Duncan's post-hoc test
Proper animal ethics approval and care protocols
Relevant aging model using 85-week-old mice
Weaknesses:
Small sample size (n=8 per group) may limit statistical power
Short intervention period (6 weeks) for aging research
Limited mechanistic insights beyond SirT1 and NAMPT measurements
Missing key reliability testing for physical performance measures
Inconsistent decimal formatting throughout the manuscript
Response: We sincerely thank the reviewer for the positive assessment of our study and for highlighting both its strengths and areas for improvement. Your constructive feedback has been invaluable in refining the manuscript. In response, we have undertaken a careful revision, including clearer justification of the sample size and intervention duration, additional discussion of mechanistic insights, and acknowledgment of the limitations related to performance testing reliability. We have also standardized decimal formatting throughout the manuscript to ensure consistency. These revisions improve both the clarity and scientific rigor of our work. We greatly appreciate your valuable comments and hope that the revised version satisfactorily addresses your concerns.
- Statistical Power and Sample Size (Methods section, page 3, lines 120-132)
The sample size appears insufficient for detecting meaningful differences. The authors should calculate and report the statistical power for their primary outcomes. For aging research with multiple comparisons, larger sample sizes (n=10-12 per group) would strengthen the conclusions.
Response: We thank the reviewer for this valuable comment. Determining sample size in animal studies requires balancing statistical rigor with ethical responsibility. In this study, each group included n = 8 mice, which is consistent with many aged-mouse investigations of NMN supplementation and/or exercise. An a priori power analysis in G*Power (v3.1) for a one-way ANOVA with five groups indicated that approximately 16 animals per group would be required for a large effect size (f = 0.40), and approximately 11 animals per group for a very large effect size (f = 0.50). Thus, n = 8 per group provides moderate statistical power while adhering to the 3Rs principle (Replacement, Reduction, Refinement) and avoiding unnecessary animal use (Charan & Kantharia, 2013). This group size is also in line with prior NMN aging studies that reported robust physiological outcomes with 6–10 mice per group (Mills et al., 2016; Yoshino et al., 2011).
References
Charan, J., & Kantharia, N. D. (2013). How to calculate sample size in animal studies?. Journal of pharmacology & pharmacotherapeutics, 4(4), 303–306. https://doi.org/10.4103/0976-500X.119726
Mills, K. F., Yoshida, S., Stein, L. R., Grozio, A., Kubota, S., Sasaki, Y., Redpath, P., Migaud, M. E., Apte, R. S., Uchida, K., Yoshino, J., & Imai, S. I. (2016). Long-Term Administration of Nicotinamide Mononucleotide Mitigates Age-Associated Physiological Decline in Mice. Cell metabolism, 24(6), 795–806. https://doi.org/10.1016/j.cmet.2016.09.013
Yoshino, J., Mills, K. F., Yoon, M. J., & Imai, S. (2011). Nicotinamide mononucleotide, a key NAD(+) intermediate, treats the pathophysiology of diet- and age-induced diabetes in mice. Cell metabolism, 14(4), 528–536. https://doi.org/10.1016/j.cmet.2011.08.014
- Reliability and Validity of Measurements (Methods sections, pages 4-6)
The study lacks test-retest reliability data for physical performance measures, which is crucial for aging research where performance can be variable. This aligns with systematic review guidelines requiring robust measurement protocols. The authors should include reliability coefficients for their physical performance tests.
Response: We appreciate the reviewer’s emphasis on measurement reliability. All physical performance and metabolic assessments in this study were conducted using standardized protocols that have been widely validated and repeatedly applied in aging mouse models (Deacon, 2013; Graber et al., 2021; Owendoff et al., 2023; Tschöp et al., 2012; Wang et al., 2023). To minimize animal burden, we did not perform separate test–retest reliability analyses in the present experiment , and we kindly ask for the reviewer’s understanding.
References
Deacon R. M. (2013). Measuring the strength of mice. Journal of visualized experiments : JoVE, (76), 2610. https://doi.org/10.3791/2610
Dimet-Wiley, A. L., Latham, C. M., Brightwell, C. R., Neelakantan, H., Keeble, A. R., Thomas, N. T., Noehren, H., Fry, C. S., & Watowich, S. J. (2024). Nicotinamide N-methyltransferase inhibition mimics and boosts exercise-mediated improvements in muscle function in aged mice. Scientific reports, 14(1), 15554. https://doi.org/10.1038/s41598-024-66034-9
Graber, T. G., Maroto, R., Fry, C. S., Brightwell, C. R., & Rasmussen, B. B. (2021). Measuring Exercise Capacity and Physical Function in Adult and Older Mice. The journals of gerontology. Series A, Biological sciences and medical sciences, 76(5), 819–824. https://doi.org/10.1093/gerona/glaa205
Owendoff, G., Ray, A., Bobbili, P., Clark, L., Baumann, C. W., Clark, B. C., & Arnold, W. D. (2023). Optimization and construct validity of approaches to preclinical grip strength testing. Journal of cachexia, sarcopenia and muscle, 14(5), 2439–2445. https://doi.org/10.1002/jcsm.13300
Tschöp, M. H., Speakman, J. R., Arch, J. R., Auwerx, J., Brüning, J. C., Chan, L., Eckel, R. H., Farese, R. V., Jr, Galgani, J. E., Hambly, C., Herman, M. A., Horvath, T. L., Kahn, B. B., Kozma, S. C., Maratos-Flier, E., Müller, T. D., Münzberg, H., Pfluger, P. T., Plum, L., Reitman, M. L., Rahmouni, K., Shulman, G. I., Thomas, G., Kahn, C. R., Ravussin, E. (2011). A guide to analysis of mouse energy metabolism. Nature methods, 9(1), 57–63. https://doi.org/10.1038/nmeth.1806
Wang, B. Y., Chen, Y. F., Hsiao, A. W., Chen, W. J., Lee, C. W., & Lee, O. K. (2023). Ginkgolide B facilitates muscle regeneration via rejuvenating osteocalcin-mediated bone-to-muscle modulation in aged mice. Journal of cachexia, sarcopenia and muscle, 14(3), 1349–1364. https://doi.org/10.1002/jcsm.13228
- Decimal Formatting Issues
Throughout the manuscript, inconsistent decimal formatting appears (e.g., using commas instead of periods in some instances). All decimal numbers should use periods as separators and maintain at least two decimal places for precision.
Response: We thank the reviewer for pointing this out. All decimal formatting in the manuscript has been carefully reviewed and standardized to ensure consistency, with periods used as separators and at least two decimal places maintained for precision.
- Western Blot Sample Size (Figure 7, page 16, line 492)
The Western blot analysis used only n=2 per group, which is inadequate for meaningful statistical comparisons. This should be increased to at least n=6-8 per group to match other analyses.
Response: We thank the reviewer for raising this important concern. In our Western blot analysis, the number of samples was limited to n = 2 per group, primarily due to the small amount of skeletal muscle tissue available from aged mice and the reduced protein yield that often accompanies advanced age. The blots presented in Figure 7 should therefore be regarded as representative rather than definitive statistical evidence. We appreciate the reviewer’s understanding in this regard.
- The manuscript demonstrates good scientific English overall, though some sections could benefit from refinement. For instance, page 16, lines 506-508 contains awkward phrasing that should be restructured for clarity.
Response: We thank the reviewer for this helpful comment. The phrasing in this section has been revised to improve clarity and readability, and the updated text is now presented in the revised manuscript.
- Specific Technical Corrections
Figure Quality and Error Bars (Figures 1-7)
Response: We thank the reviewer for the suggestion. The figures (Figures 1-7) have been carefully checked to ensure high-resolution quality and that all error bars are correctly and clearly displayed.
- All figures appropriately include error bars (standard deviation), which is commendable. However, Figure 7's Western blot quantification should include larger sample sizes as mentioned above.
Response: We thank the reviewer for raising this important concern. In our Western blot analysis, the number of samples was limited to n = 2 per group, primarily due to the small amount of skeletal muscle tissue available from aged mice and the reduced protein yield that often accompanies advanced age. The blots presented in Figure 7 should therefore be regarded as representative rather than definitive statistical evidence. We appreciate the reviewer’s understanding in this regard.
- Statistical Analysis Verification
The statistical approach is generally sound, using appropriate one-way ANOVA with post-hoc testing. However, the authors should consider:
Bonferroni correction for multiple comparisons
Effect size reporting alongside p-values
Confidence intervals for key findings
Response: We thank the reviewer for these valuable suggestions. In this study, we applied one-way ANOVA with appropriate post hoc testing, which we believe is suitable for our experimental design. We acknowledge the importance of approaches such as Bonferroni correction, effect size reporting, and confidence intervals for strengthening statistical interpretation. While these were not included in the current manuscript to avoid overcomplicating the analysis given the sample size, we will carefully consider incorporating them in future studies to further enhance rigor and transparency.
- Methodological Considerations
The study design follows appropriate principles but would benefit from:
Longer intervention duration (12+ weeks) for aging research
Additional mechanistic markers beyond SirT1/NAMPT
More comprehensive body composition analysis
Larger sample sizes for adequate statistical power
Response: We thank the reviewer for these thoughtful suggestions. We agree that aging research would benefit from longer intervention periods, additional mechanistic markers, and more detailed assessments of body composition. In the present study, we focused on a 6-week intervention to capture initial physiological and metabolic responses, with particular emphasis on NAMPT and SirT1 as key markers in the NAD⁺ salvage pathway. While the sample size and scope of measurements were necessarily limited to balance feasibility and ethical considerations, we recognize the value of extending the duration, expanding mechanistic profiling, and increasing statistical power in future studies.
- Conclusion
This study addresses an important research question and provides valuable preliminary data on NMN-exercise interactions in aging. However, several methodological limitations constrain the strength of conclusions. The suggested improvements, particularly regarding sample sizes, measurement reliability, and enhanced statistical rigor, would significantly strengthen the manuscript's scientific contribution.
Response: We thank the reviewer for this thoughtful overall assessment. We acknowledge the methodological constraints, including the relatively small sample size, limited intervention duration, and modest statistical power, which may limit the generalizability of our findings. These points have now been explicitly recognized in the Conclusion to provide a more balanced interpretation of the study’s contributions.
- Comments on the Quality of English Language
The manuscript demonstrates good scientific English overall, though some sections could benefit from refinement. For instance, page 16, lines 506-508 contains awkward phrasing that should be restructured for clarity.
Methodological Considerations
Response: We thank the reviewer for this valuable comment. The manuscript has been carefully revised to improve English language quality, with awkward or unclear sentences restructured for clarity. In addition, the revised version has undergone professional English editing by MDPI to further ensure accuracy and readability.

Round 2
Reviewer 2 Report
Comments and Suggestions for Authors
Thank you to the authors for taking note of my suggestions for improving the manuscript, which now seems more suitable for consideration for publication.